# Expressive Yet Tractable Bayesian Deep Learning via Subnetwork Inference

## Abstract

The Bayesian paradigm has the potential to solve some of the core issues in modern deep learning, such as poor calibration, data inefficiency, and catastrophic forgetting. However, scaling Bayesian inference to the high-dimensional parameter spaces of deep neural networks requires restrictive approximations. In this paper, we propose performing inference over only a small subset of the model parameters while keeping all others as point estimates. This enables us to use expressive posterior approximations that would otherwise be intractable for the full model. In particular, we develop a practical and scalable Bayesian deep learning method that first trains a point estimate, and then infers a full covariance Gaussian posterior approximation over a subnetwork. We propose a subnetwork selection procedure which aims to maximally preserve posterior uncertainty. We empirically demonstrate the effectiveness of our approach compared to point-estimated networks and methods that use less expressive posterior approximations over the full network.

## 1 Introduction

Deep neural networks (DNNs) still suffer from critical shortcomings that make them unfit for important applications. For instance, DNNs tend to be *poorly calibrated and overconfident* in their predictions, especially when there is a shift in the train and test distributions (Nguyen et al., 2015; Guo et al., 2017). To reliably inform decision making, DNNs must be able to robustly quantify the *uncertainty* in their predictions, which is particularly important in safety-critical areas such as healthcare or autonomous driving (Amodei et al., 2016; Filos et al., 2019a; Fridman et al., 2019).

Bayesian modeling (Ghahramani, 2015; Gal, 2016) presents a principled way to capture predictive uncertainty via the posterior distribution over model parameters. Unfortunately, due to their non-linearities, exact posterior inference is intractable in DNNs. Despite recent successes in the field of Bayesian deep learning (Blundell et al., 2015; Gal & Ghahramani, 2016; Osawa et al., 2019; Maddox et al., 2019; Dusenberry et al., 2020), existing methods are only made scalable to modern DNNs with large numbers of parameters by invoking unrealistic assumptions. This severely limits the expressiveness of the inferred posterior and thus deteriorates the quality of the induced uncertainty estimates (Ovadia et al., 2019; Fort et al., 2019; Foong et al., 2019a; Ashukha et al., 2020a).

Due to the heavy overparameterization of DNNs, their accuracy is well-preserved by a small subnetwork (Cheng et al., 2017). Additionally, recent work by Izmailov et al. (2019) has shown how performing inference over a low dimensional subspace of the weights can result in accurate uncertainty quantification. These observations prompt the following question for a DNN's uncertainty: *Can a full DNN's model uncertainty be well-preserved by a small subnetwork's model uncertainty?* We answer this question in the affirmative. We show both theoretically and empirically that the full network posterior can be well represented by a subnetwork's posterior. As a result, we can use more expensive but faithful posterior approximations over just that subnetwork. We show that this achieves better uncertainty quantification than if we use cheaper, but more crude, posterior approximations over the full network.

The contributions of this paper are as follows:

1. We propose a new Bayesian deep learning approach that performs Bayesian inference over only a small subset of the model weights and keeps all other weights deterministic. This allows us to use expressive posterior approximations that are typically intractable in DNNs.

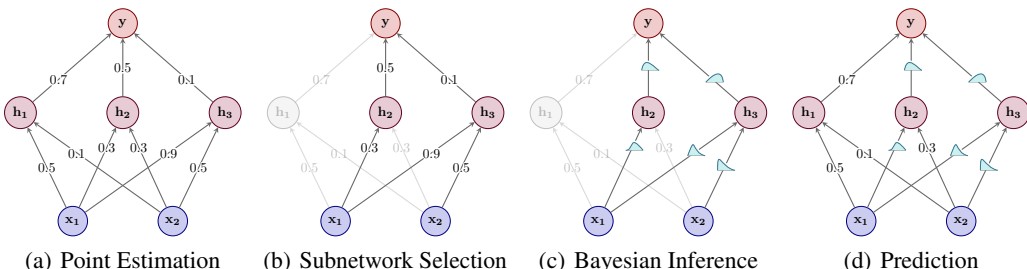

| (a) Point Estimation | (b) Subnetwork Selection | (c) Bayesian Inference | (d) Prediction |

Figure 1: Schematic illustration of our proposed approach. (a) We train a neural network using standard techniques to obtain a point estimate of the weights. (b) We identify a small subset of the weights. (c) We estimate a posterior distribution over the selected subnetwork via Bayesian inference techniques. (d) We make predictions using the full network of mixed Bayesian/deterministic weights.

2. As a concrete instantiation of this framework, we develop a practical and scalable Bayesian deep learning method that uses the linearized Laplace approximation to infer a full-covariance Gaussian posterior over a subnetwork within a point-estimated neural network.

3. We formally characterize the discrepancy between the posterior distributions over a subnetwork and the full network (in terms of their Wasserstein distance) in the linearized model, and derive a theoretically motivated strategy to select a subnetwork that minimizes this discrepancy under certain assumptions.

4. We empirically show, on various benchmarks, that our method compares favourably against point-estimated networks and other Bayesian deep learning methods, experimentally confirming that expressive subnetwork inference is superior to crude inference over full networks.

## 2    SUBNETWORK POSTERIOR APPROXIMATION

Bayesian neural networks (BNNs) aim to capture *model uncertainty*, i.e., uncertainty about the choice of weights $W$ which arises due to multiple plausible explanations of the training data $\{y, X\}$. Here, $y$ is the dependent variable (e.g. classification label) and $X$ is the feature matrix. A prior distribution $p(\mathbf{W})$ is specified over the BNN's weights. We wish to infer their full *posterior distribution*

$$p(\mathbf{W}|\boldsymbol{y}, \boldsymbol{X}) \quad \propto \quad p(\boldsymbol{y}|\boldsymbol{X}, \mathbf{W})\, p(\mathbf{W}) \,. \tag{1}$$

To make predictions, we then estimate the *posterior predictive* distribution that averages the network's predictions across all possible settings of the weights, weighted by their posterior probability, i.e.

$$p(\mathbf{y}^*|\boldsymbol{X}^*, \boldsymbol{y}, \boldsymbol{X}) = \int_{\mathbf{W}} p(\mathbf{y}^*|\boldsymbol{X}^*, \mathbf{W}) p(\mathbf{W}|\boldsymbol{y}, \boldsymbol{X}) d\mathbf{W} \,. \tag{2}$$

Unfortunately, due to the size of modern deep neural networks, it is not only intractable to infer the exact posterior distribution $p(\mathbf{W}|\boldsymbol{y}, \boldsymbol{X})$ in Eq. (1), but it is even computationally challenging to properly approximate it. As a consequence, crude posterior approximations such as complete factorization are commonly employed (Blundell et al., 2015; Hernández-Lobato & Adams, 2015; Kingma et al., 2015; Khan et al., 2018; Osawa et al., 2019), i.e. $p(\mathbf{W}|\boldsymbol{y}, \boldsymbol{X}) \approx \prod_{d=1}^{D} q(\mathrm{w}_d)$ where $\mathrm{w}_d$ denotes the $d$-th weight in the $D$-dimensional neural network weight vector $\mathbf{W} \in \mathbb{R}^D$ (the concatenation and flattening of all layers' weight matrices). Clearly, this is a very wishful assumption; In practise, it suffers from severe pathologies (Foong et al., 2019a;b).

In this work, we question the implicit assumption that a good posterior approximation needs to include *all* BNN parameters. Instead, we aim to perform inference only over a *small subset* of the weights. This approach is well-motivated for two reasons:

1. **Overparameterization:** Maddox et al. (2020) have shown that, in the neighborhood of local optima, there are many directions that leave the NN's predictions unchanged. Moreover, NNs can be heavily pruned without sacrificing test-set accuracy (Frankle & Carbin, 2019). Thus, the majority of a NN's predictive power might be isolated to a small subnetwork.

2. **Inference over submodels:** Previous work[1] has provided evidence that inference can be effective even when not done on the full parameter space. Izmailov et al. (2019) performed inference over a low-dimensional projection of the weights. Neural-linear models, which give a Bayesian treatment to only the last layer of a DNN, have shown to be competitive with full-network approaches (Riquelme et al., 2018; Ober & Rasmussen, 2019).

We thus combine these ideas, making the following two-step approximation of the posterior in Eq. (1):

$$p(\mathbf{W}|\boldsymbol{y}, \boldsymbol{X}) \;\approx\; p(\mathbf{W}_S|\boldsymbol{y}, \boldsymbol{X}) \prod_r \delta(\mathrm{w}_r - \mathrm{w}_r^*) \;\approx\; q(\mathbf{W}_S) \prod_r \delta(\mathrm{w}_r - \mathrm{w}_r^*) \,. \tag{3}$$

The first approximation decomposes the full neural network posterior $p(\mathbf{W}|\boldsymbol{y}, \boldsymbol{X})$ into a posterior $p(\mathbf{W}_S|\boldsymbol{y}, \boldsymbol{X})$ over the subnetwork $\mathbf{W}_S$ and delta functions $\delta(\mathrm{w}_r - \mathrm{w}_r^*)$ over all remaining weights $\{\mathrm{w}_r\}_r$, keeping them at fixed values $\mathrm{w}_r^* \in \mathbb{R}$. This can be viewed as *pruning the variances* of the weights $\{\mathrm{w}_r\}_r$ to zero, which is in contrast to ordinary weight pruning methods that set the *weights* $\{\mathrm{w}_r\}_r$ themselves to zero. The second approximation is a result of posterior inference over the subnetwork still being intractable. In turn, we introduce the approximate distribution $q(\mathbf{W}_S)$. Yet, as the subnetwork is much smaller than the full network, we can afford to make $q(\mathbf{W}_S)$ expressive and able to capture rich dependencies across the weights within the subnetwork.

## 3  SUBNETWORK INFERENCE VIA LAPLACE APPROXIMATION

To obtain a method that is as practical as possible, we propose to use inference techniques that can estimate a posterior distribution *post-hoc* from a point-estimated network. The *Laplace approximation* (MacKay, 1992) is well-suited to this task as it derives the approximate posterior from the local optimization landscape. Other inference procedures, such as SWAG (Maddox et al., 2019), could also be used. Nevertheless, we focus on Laplace due to it being a well-studied, fundamental technique.

**Step #1: Point Estimation.**   The first step of the procedure is to train a neural network to obtain a point estimate of the weights, denoted $\boldsymbol{W}_{MAP}$. This estimate should respect the Bayesian model given in Eq. (1), and therefore we optimize the *maximum a-posteriori* (MAP) objective:

$$\boldsymbol{W}_{MAP} = \arg\max_{\boldsymbol{W}} \left[ \log p(\mathbf{y}|\boldsymbol{X}, \boldsymbol{W}) + \log p(\boldsymbol{W}) \right] \,. \tag{4}$$

This can be done using standard stochastic gradient-based optimization methods commonly-used in modern deep learning (Goodfellow et al., 2016). This step is illustrated in Fig. 1 (a).

**Step #2: Subnetwork Selection.**   The second step is to identify a small subnetwork $\boldsymbol{W}_S$. Ideally, we would like to identify the subnetwork whose posterior is 'closest' to the full-network posterior. We formalize this argument in Section 4 and describe a principled strategy that, under certain conditions, minimizes the 2-Wasserstein distance between the sub- and full-network posteriors. All other weights not belonging to that subnetwork are then assigned fixed values: the MAP estimates obtained in Step #1. See Fig. 1 (b) for an illustration of this step.

**Step #3: Bayesian Inference.**   Given the subnetwork point estimate $\boldsymbol{W}_{MAP}^S$, we use the Laplace approximation to infer a full-covariance Gaussian posterior distribution over the subnetwork $\mathbf{W}_S$:

$$p(\mathbf{W}_S|\boldsymbol{y}, \boldsymbol{X}) \;\approx\; q(\mathbf{W}_S) \;=\; \mathcal{N}\left(\mathbf{W}_S; \boldsymbol{W}_{MAP}^S, H^{-1}\right) \tag{5}$$

where the posterior covariance matrix $H^{-1} \in \mathbb{R}^{D \times D}$ corresponds to the inverse of the average Hessian of the negative log posterior, i.e. $H = N\mathbb{E}\left[-\partial^2 \log p(\mathbf{y}|\boldsymbol{X}, \boldsymbol{W})/\partial \boldsymbol{W}^2\right] + \lambda \boldsymbol{I}$. Here, the expectation is w.r.t. the data generating distribution and $\lambda$ is the precision of a zero-mean factorized Gaussian prior $p(\mathbf{W}) = \mathcal{N}(\mathbf{W}; \mathbf{0}, \lambda^{-1}\boldsymbol{I})$. In practice, we approximate the Hessian $H$ with the *generalized Gauss-Newton (GGN) matrix* $\widetilde{H}$ (Schraudolph, 2002), i.e.

$$\widetilde{H} = \sum_{n=1}^{N} \boldsymbol{J}_n^\top \boldsymbol{H}_n \boldsymbol{J}_n + \lambda \boldsymbol{I}, \;\; \text{with} \;\; \boldsymbol{J}_n = \frac{\partial \boldsymbol{f}(\boldsymbol{x}_n, \boldsymbol{W})}{\partial \boldsymbol{W}} \;\; \text{and} \;\; \boldsymbol{H}_n = \frac{\partial^2 L(\boldsymbol{y}_n, \boldsymbol{f}(\boldsymbol{x}_n, \boldsymbol{W}))}{\partial^2 \boldsymbol{f}(\boldsymbol{x}_n, \boldsymbol{W})} \tag{6}$$

---

[1]See Section 6 for a more thorough discussion of related work.

where $\boldsymbol{J}_n \in \mathbb{R}^{O \times D}$ is the Jacobian of the neural network features $\boldsymbol{f}(\boldsymbol{x}_n, \boldsymbol{W}) \in \mathbb{R}^O$ w.r.t. the weights $\boldsymbol{W}$, and $\boldsymbol{H}_n \in \mathbb{R}^{O \times O}$ is the Hessian of the loss $L(\boldsymbol{y}_n, \boldsymbol{f}(\boldsymbol{x}_n, \boldsymbol{W}))$ w.r.t. the features $\boldsymbol{f}(\boldsymbol{x}_n, \boldsymbol{W})$. The GGN $\widetilde{H}$ has clear practical advantages over the Hessian $H$; see Martens & Sutskever (2011) and Martens (2016). Using the Laplace approximation with the GGN Hessian can be viewed as an implicit *local linearization* of the underlying neural network $\boldsymbol{f}(\boldsymbol{x}, \boldsymbol{W})$ at its MAP estimate $\boldsymbol{W}_{MAP}$,

$$\boldsymbol{f}_{lin}^{MAP}(\boldsymbol{x}, \boldsymbol{W}) = \boldsymbol{f}(\boldsymbol{x}, \boldsymbol{W}_{MAP}) + \boldsymbol{J}_{\boldsymbol{W}_{MAP}}(\boldsymbol{x})(\boldsymbol{W} - \boldsymbol{W}_{MAP}) \tag{7}$$

where $\boldsymbol{J}_{\boldsymbol{W}_{MAP}}(\boldsymbol{x}) = \partial \boldsymbol{f}(\boldsymbol{x}, \boldsymbol{W}_{MAP})/\partial \boldsymbol{W}_{MAP} \in \mathbb{R}^{O \times D}$ (Immer et al., 2020). Note that the model in Eq. (7) is *linear* in $\boldsymbol{W}$, as only the term $\boldsymbol{J}_{\boldsymbol{W}_{MAP}}(\boldsymbol{x})\boldsymbol{W}$ depends linearly on $\boldsymbol{W}$, while the other terms are constant w.r.t. $\boldsymbol{W}$ and can thus be subsumed into an additive bias term (Khan et al., 2019). The GGN approximation thus locally turns the underlying probabilistic model from a Bayesian neural network into a (generalized) linear model, with basis function expansion $\boldsymbol{J}_{\boldsymbol{W}_{MAP}}(\boldsymbol{x})$ of covariate $\boldsymbol{x}$ (Immer et al., 2020). Put differently, linearized Laplace in the neural network $\boldsymbol{f}(\boldsymbol{x}, \boldsymbol{W})$ is *equivalent* to ordinary Laplace in the linear model $\boldsymbol{f}_{lin}^{MAP}(\boldsymbol{x}, \boldsymbol{W})$ in Eq. (7), as the GGN $\widetilde{H}$ corresponding to $\boldsymbol{f}(\boldsymbol{x}, \boldsymbol{W})$ in Eq. (6) is equivalent to the Hessian $H$ corresponding to $\boldsymbol{f}_{lin}^{MAP}(\boldsymbol{x}, \boldsymbol{W})$ in Eq. (7) (Khan et al., 2019). This is a useful property that will allow us to derive a principled subnetwork selection strategy in Section 4. This step is illustrated in Fig. 1 (c). We emphasize that this whole procedure (i.e. Steps #1-#3) is a perfectly valid mixed inference strategy, performing full Laplace inference over the selected subnetwork and MAP inference over all remaining weights.

**Step #4: Prediction.**  Given the linearized Laplace approximation over the subnetwork $\mathbf{W}_S$ in Eqs. (5) and (6), i.e. $q(\mathbf{W}_S) = \mathcal{N}(\mathbf{W}_S; \boldsymbol{W}_{MAP}^S, \widetilde{H}^{-1})$, we can then compute the posterior predictive distribution. While, traditionally, one would compute the predictive distribution using the original Bayesian neural network likelihood, i.e. $p(\mathbf{y}|\boldsymbol{X}, \mathbf{W}) = p(\mathbf{y}|\boldsymbol{f}(\boldsymbol{x}, \boldsymbol{W}))$, Immer et al. (2020) recently suggested that, since inference was (implicitly) done in the GGN-linearized model, it is more principled to instead predict using the linearized likelihood Eq. (7), i.e. $p(\mathbf{y}|\boldsymbol{X}, \mathbf{W}) = p(\mathbf{y}|\boldsymbol{f}_{lin}^{MAP}(\boldsymbol{x}, \boldsymbol{W}))$. This provides a formal justification for the empirical superiority of this approach observed previously (Lawrence, 2001; Foong et al., 2019b). We thus obtain the *linearized predictive distribution*

$$p(\mathbf{y}^*|\boldsymbol{X}^*, \boldsymbol{y}, \boldsymbol{X}) \approx \int_{\mathbf{W}} p(\mathbf{y}^*|\boldsymbol{f}_{lin}^{MAP}(\boldsymbol{X}^*, \boldsymbol{W})) \, \mathcal{N}(\mathbf{W}_S; \boldsymbol{W}_{MAP}^S, \widetilde{H}^{-1}) \prod_r \delta(\mathrm{w}_r - \mathrm{w}_r^*) \, d\mathbf{W} \,. \tag{8}$$

There are two ways to compute Eq. (8): Firstly, via a Monte Carlo approximation $p(\mathbf{y}^*|\boldsymbol{X}^*, \boldsymbol{y}, \boldsymbol{X}) \simeq \frac{1}{M} \sum_{m=1}^M p(\mathbf{y}^*|\boldsymbol{f}_{lin}^{MAP}(\boldsymbol{X}^*, \widetilde{\boldsymbol{W}}_m))$ by sampling $\widetilde{\boldsymbol{W}}_m$ from $\mathcal{N}(\boldsymbol{W}_{MAP}^S, \widetilde{H}^{-1})$ and $\prod_r \delta(\mathrm{w}_r - \mathrm{w}_r^*)$, the latter of which is trivial. Secondly, due to linearity of $p(\mathbf{y}^*|\boldsymbol{f}_{lin}^{MAP}(\boldsymbol{X}^*, \boldsymbol{W}))$, there are closed-form expressions which are exact for Gaussian likelihoods (i.e. regression) and approximate for categorical ones (i.e. classification) (Bishop, 2006; Gibbs, 1998). This step is illustrated in Fig. 1 (d).

## 4 PRINCIPLED SUBNETWORK SELECTION FOR LINEAR(IZED) MODELS

We next analyze the subnetwork inference procedure described in Section 3 for the case of a *generalized linear model* (GLM) (Nelder & Baker, 1972), which models the expected response $y_n$ given the basis function expansion of the covariates $\boldsymbol{\phi}_n = \boldsymbol{\phi}(\boldsymbol{x}_n)$ as

$$\mathbb{E}[y_n|\boldsymbol{\phi}_n] = g^{-1}(\mathbf{w}^T \boldsymbol{\phi}_n). \tag{9}$$

Here, $\mathbf{w} \in \mathbb{R}^D$ is the vector of model parameters (which subsumes a scalar bias $\beta_0$ for notational convenience) and $g^{-1}(\cdot)$ denotes a *link function* such that $g^{-1} : \mathbb{R} \mapsto \mu_{y|\boldsymbol{\phi}}$. In particular, we consider a *Bayesian* GLM, by specifying a prior distribution $p(\mathbf{w})$ over model parameters and aiming to infer the posterior distribution $p(\mathbf{w}|\mathbf{y}, \boldsymbol{\Phi}) \propto p(\mathbf{y}|\boldsymbol{\Phi}, \mathbf{w})p(\mathbf{w})$, where $\boldsymbol{\Phi} = [\boldsymbol{\phi}_1, \dots \boldsymbol{\phi}_N]^T$.

1. **Point Estimation.** Obtain the MAP estimate, $\mathbf{w}_{MAP} = \arg\max_{\mathbf{w}} \log p(\boldsymbol{y}|\boldsymbol{\Phi}, \mathbf{w}) + \log p(\mathbf{w})$. For commonly-used link functions (e.g. the identity in case of a Gaussian likelihood for regression, or the sigmoid/softmax function in case of a categorical likelihood for classification) and commonly-used priors (e.g. a Gaussian), the log-posterior $\propto \log p(\boldsymbol{y}|\boldsymbol{\Phi}, \mathbf{w}) + \log p(\mathbf{w})$ is concave. This allows for simple gradient-based MAP optimisation. It also makes a full-covariance Gaussian, estimated via Laplace, a faithful approximation to the true, uni-modal posterior, i.e.

$$p(\mathbf{w}|\mathbf{y}, \boldsymbol{\Phi}) \approx \widetilde{p}(\mathbf{w}|\mathbf{y}, \boldsymbol{\Phi}) = \mathcal{N}(\mathbf{w}; \mathbf{w}_{MAP}, H^{-1}) \tag{10}$$

where $H$ is the Hessian defined in Section 3. Note that for the GLM we consider, the Hessian $H$ is equivalent to the GGN $\widetilde{H}$ defined in Eq. (6), meaning that an *ordinary* Laplace approximation is equivalent to a *linearized* Laplace approximation (Martens, 2016). For the case of an identity link function (i.e. a Gaussian likelihood with noise variance $\sigma_0^2$) and a Gaussian prior $\mathbf{w} \sim \mathcal{N}(\mathbf{0}, \boldsymbol{\Lambda}_0^{-1})$, the MAP estimate even has a closed-form expression, $\mathbf{w}_{MAP} = (\boldsymbol{\Phi}^T\boldsymbol{\Phi} + \sigma_0^2\boldsymbol{\Lambda}_0)^{-1}\boldsymbol{\Phi}^T\boldsymbol{y}$. Here, the Laplace approximation in Eq. (10) *exactly corresponds to the true posterior*, i.e. $\widetilde{p}(\mathbf{w}|\mathbf{y}, \boldsymbol{\Phi}) = p(\mathbf{w}|\mathbf{y}, \boldsymbol{\Phi})$. We will thus refer to the posterior $\widetilde{p}(\mathbf{w}|\mathbf{y}, \boldsymbol{\Phi})$ in Eq. (10) as the *full posterior*.

2. **Subnetwork Selection.** Select a subset of $S$ model weights via a method of choice, yielding a binary vector $\boldsymbol{m} \in \mathbb{R}^D$ where $m_d = 1$ if the $d$-th weight is part of the subset, and $m_d = 0$ otherwise. For convenience, we define the binary mask matrix $\boldsymbol{M}_S = \boldsymbol{m}\boldsymbol{m}^\top \in \mathbb{R}^{D \times D}$ which contains 1s in the rows/columns corresponding to the $S$ subnetwork weights[2], and 0s otherwise.

3. **Bayesian Inference.** Compute the posterior over the subnetwork via a Laplace approximation:
$$p_S(\mathbf{w}|\boldsymbol{y}, \boldsymbol{\Phi}) = \mathcal{N}(\mathbf{w}; \mathbf{w}_{MAP}, \boldsymbol{M}_S \odot H^{-1}) . \tag{11}$$
Firstly, note that the *mean* of the subnetwork posterior in Eq. (11) is the MAP estimate $\mathbf{w}_{MAP}$ and thus equal to the mean of the full posterior $\widetilde{p}(\mathbf{w}|\mathbf{y}, \boldsymbol{\Phi})$ in Eq. (10). Secondly, note that the *covariance matrix* of the subnetwork posterior in Eq. (11) is the element-wise product $\boldsymbol{M}_S \odot H^{-1}$, which masks the (co-)variances of all weights *not* belonging to the subnetwork to zero, effectively making them deterministic. More precisely, the subnetwork covariance matrix, $\boldsymbol{M}_S \odot H^{-1}$, is a $D \times D$ matrix that is equal to the full posterior covariance matrix $H^{-1}$ in the rows/columns of the $S$ weights in the subnetwork, and zero in the rows/columns of all other $D - S$ weights.

We consider what we term the *posterior gap*—the Wasserstein distance[3] (in particular the squared 2-Wasserstein distance) between the posterior distribution over the full network and the posterior distribution over the subnetwork. The proofs for all results below will be presented in Appendix A.

**Proposition 1** (Posterior Gap). *For a subnetwork of size $S < D$, the Wasserstein gap between the full posterior $\widetilde{p}(\mathbf{w}|\boldsymbol{y}, \boldsymbol{\Phi})$ in Eq. (10) and the subnetwork posterior $p_S(\mathbf{w}|\boldsymbol{y}, \boldsymbol{\Phi})$ in Eq. (11) is:*
$$W[\widetilde{p}(\mathbf{w}|\boldsymbol{y}, \boldsymbol{\Phi}) \,||\, p_S(\mathbf{w}|\boldsymbol{y}, \boldsymbol{\Phi})] = \textstyle\sum_{d=1}^{D} (1 + m_{dd})\,\sigma_d^2 - \mathrm{trace}(2(H^{-1}(\boldsymbol{M}_S \odot H^{-1}))^{1/2}) . \tag{12}$$

The optimal subnetwork should then minimize the posterior gap in Eq. (12). However, for full covariance matrices $H^{-1}$ and a large number of weights $D$, this will generally be infeasible as Eq. (12) depends on *all* entries of the $D \times D$-matrix $H^{-1}$, which is intractable to compute/store. To derive a practical subnetwork selection strategy, we assume the covariance matrix to be diagonal.

**Corollary 1.1** (Optimality of Maximum Variance Subnetwork Selection under Decorrelation). *For a generalized linear model with posterior covariance matrix $H^{-1} = diag(\sigma_1^2, \ldots, \sigma_D^2)$, the optimal subnetwork under the Wasserstein gap is comprised of the $S$ weights with the largest variances $\sigma_d^2$.*

Finally, since a GGN-linearized neural network, as in Eq. (7), corresponds to a GLM with basis expansion $\boldsymbol{\phi}_n = \boldsymbol{J}_{\boldsymbol{W}_{MAP}}(\boldsymbol{x}_n) = \partial \boldsymbol{f}(\boldsymbol{x}_n, \boldsymbol{W}_{MAP})/\partial \boldsymbol{W}_{MAP}$ (see Step #3 in Section 3), Corollary 1.1 implies that under decorrelation, the optimal subnetwork comprises of the weights with the largest variances. In practice, even just computing the diagonal of the covariance matrix is challenging, so we use a diagonal Laplace approximation which instead computes the inverse of the diagonal of the GGN (see e.g. Ritter et al. (2018)). Finally, note that we only have to make the decorrelation assumption for the purposes of subnetwork selection – when doing posterior inference over the selected subnetwork, *we estimate a full covariance matrix* for maximal expressiveness, as described in Step #3 in Section 3. In our experiments in Section 5, we empirically show that making the decorrelation assumption for subnetwork selection but then using a full-covariance Gaussian for inference performs significantly better than directly making the decorrelation assumption for inference (e.g. mean-field variational inference, diagonal Laplace).

## 5 EMPIRICAL ANALYSIS

We empirically assess the effectiveness of subnetwork inference compared to point-estimated NNs and methods that do less expressive inference over the full network. We consider three tasks: 1) small-scale toy regression, 2) medium-scale tabular regression, and 3) large-scale image classification.

---

[2]For consistency, we will keep referring to the $S$ selected linear model weights as a "subnetwork".

[3]We use the Wasserstein distance instead of the more common Kullback–Leibler divergence because the Wasserstein is well-defined for degenerate distributions and is an actual distance metric (i.e. symmetric).

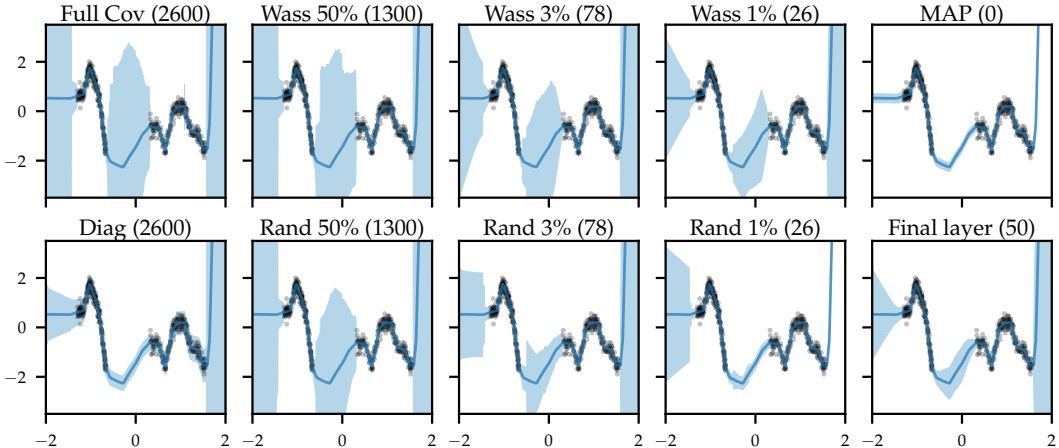

Figure 2: Predictive distributions (mean $\pm$ std) for 1D regression. The numbers in brackets denote the number of parameters over which inference was done (out of 2600 in total). Wasserstein-based subnetwork inference maintains richer predictive uncertainties at smaller parameter counts.

## 5.1 HOW DOES SUBNETWORK INFERENCE RETAIN POSTERIOR PREDICTIVE UNCERTAINTY?

We first assess how the predictive distribution of a full-covariance Gaussian posterior over a selected subnetwork qualitatively compares to that obtained from 1) a full-covariance Gaussian over the *full* network (*Full Cov*), 2) a *factorised* Gaussian posterior over the full network (*Diag*), 3) a full-covariance Gaussian over only the (*Final layer*) of the network (Kristiadi et al., 2020), and 4) a point estimate (*MAP*). For subnetwork inference, we consider both Wasserstein (*Wass*) (as described in Section 4) and uniform random subnetwork selection (*Rand*) to obtain subnetworks that comprise of only 50%, 3% and 1% of the model parameters. Note that while for this toy example, we could in principle use the full covariance matrix for the purpose of subnetwork selection, we still just use its diagonal (as described in Section 4) for consistency. Our NN consists of 2 ReLU hidden layers with 50 hidden units each. We employ a homoscedastic Gaussian likelihood function where the noise variance is optimised with maximum likelihood. We use GGN Laplace inference over network weights (not biases) in combination with the linearized predictive distribution in Eq. (8). Thus, all approaches considered share their predictive mean, allowing us to better compare their uncertainty estimates. All approaches share a single prior precision of $\lambda = 3$.

We use a synthetic 1D regression task with two separated clusters of inputs (Antorán et al., 2020), allowing us to probe for 'in-between' uncertainty (Foong et al., 2019b). Results are shown in Fig. 2. Subnetwork inference preserves more of the uncertainty of full network inference than diagonal Gaussian or final layer inference while doing inference over fewer weights. By capturing weight correlations, subnetwork inference retains uncertainty in between clusters of data. This is true for both random and Wasserstein subnetwork selection. However, the latter preserves more uncertainty with smaller subnetworks. Finally, the strong superiority to diagonal Laplace shows that making a diagonal assumption for subnetwork selection but then using a full-covariance Gaussian for inference (as we do) performs much signficantly better than making a diagonal assumption for the inferered posterior directly. These results suggest that **expressive inference over a carefully selected subnetwork retains more predictive uncertainty than crude approximations over the full network**.

## 5.2 SUBNETWORK INFERENCE IN LARGE MODELS VS FULL INFERENCE OVER SMALL MODELS

Secondly, we study the following natural question: "Why should one use subnetwork inference in a large model when one can just perform full network inference over a smaller model?" We explore this by considering 4 fully connected models of increasing size. These have numbers of hidden layers $h_d = \{1, 2\}$ and hidden layer widths $w_d = \{50, 100\}$. For a dataset with input dimension $i_d$, the number of weights is given by $D = (i_d + 1)w_d + (h_d - 1)w_d^2$. Our 2 hidden layer, 100 hidden unit models have a weight count of the order $10^4$. Full covariance inference in these models borders the limit of computational tractability on commercial hardware. We first obtain a MAP estimate of each

model's weights and our homoscedastic likelihood function's noise variance. We then perform full network GGN Laplace inference for each model. We also use our proposed Wassertein rule to prune every network's weight variances such that the number of variances that remain matches the size of every smaller network under consideration. In all cases, we employ the linearized predictive in Eq. (7). Consequently, networks with the same number of weights make the same mean predictions. Increasing the number of weight variances considered will thus only increase predictive uncertainty.

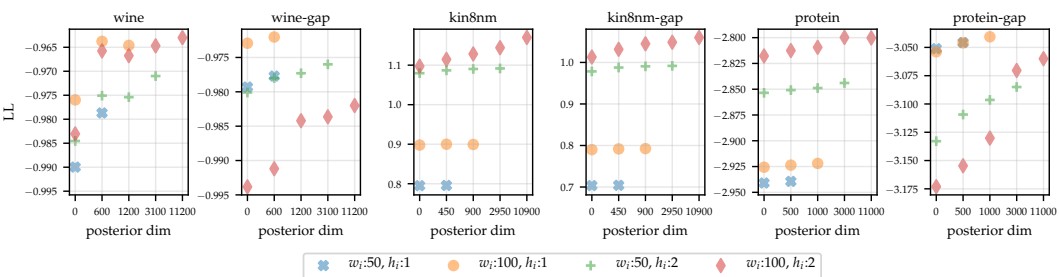

Figure 3: Mean test log-likelihood values obtained on UCI datasets across all splits. Different markers indicate models with different numbers of weights. The horizontal axis indicates the number of weights over which full covariance inference is performed. 0 corresponds to MAP parameter estimation, and the rightmost setting for each marker corresponds to full network inference.

We employ 3 tabular datasets of increasing size (input dimensionality, n. points): wine (11, 1439), kin8nm (8, 7373) and protein (9, 41157). We consider their standard train-test splits (Hernández-Lobato & Adams, 2015) and their gap variants (Foong et al., 2019b), designed to test for out-of-distribution uncertainty. Details are provided in Appendix C.4. For each split, we set aside 15% of the train data as a validation set. We use these for early stopping when finding MAP estimates and for selecting the weights' prior precision. We keep other hyperparameters fixed across all models and datasets. Results are in Fig. 3.

We present mean test log-likelihood (LL) values, as these take into account both accuracy and uncertainty. Larger models tend to perform better when doing MAP inference, with wine-gap and protein-gap being exceptions. We also find larger models improve over their respective MAP LLs more than small ones when performing approximate inference over the same numbers of weights. We conjecture this is due to an abundance of degenerate directions (weights) in the weight posterior of all models (Maddox et al., 2020). Full network inference in small models captures information about both useful and non-useful weights. In larger models, our subnetwork selection strategy allows us to dedicate a larger proportion of our resources to modelling informative weight variances and covariances. In 3 out of 6 datasets, we find abrupt increases in LL as we increase the number of weights over which we perform inference, followed by a plateau. Such plateaus might be explained by all of the most informative weight variances having already been accounted for. These results suggest that, **given the same amount of compute, larger models benefit more from subnetwork inference than small ones.**

## 5.3 SCALING TO IMAGE CLASSIFICATION WITH DISTRIBUTION SHIFT

We now assess the robustness of large convolutional neural networks with subnetwork inference to distribution shift on image classification tasks compared to the following baselines: point-estimated networks (MAP), Bayesian deep learning methods that do less expressive inference over the full network: MC Dropout (Gal & Ghahramani, 2016), diagonal Laplace, VOGN (Osawa et al., 2019) (all of which assume factorisation of the weight posterior), and SWAG (Maddox et al., 2019) (which assumes a diagonal plus low-rank posterior). We also benchmark deep ensembles (Lakshminarayanan et al., 2017). The latter is considered state-of-the-art for uncertainty quantification in deep learning (Ovadia et al., 2019; Ashukh et al., 2020a). We use ensembles of 5 DNNs, as suggested by (Ovadia et al., 2019), and 16 samples for MC Dropout, diagonal Laplace and SWAG. We use a Dropout probability of $0.1$ and a prior precision of $\lambda = 40,000$ for diagonal Laplace, found via grid search. We apply all approaches to ResNet-18 (He et al., 2016), which is composed of an input convolutional block, 8 residual blocks and a linear layer, for a total of 11,168,000 parameters. For subnetwork inference, we compute the linearized predictive distribution in Eq. (8) via the closed-form

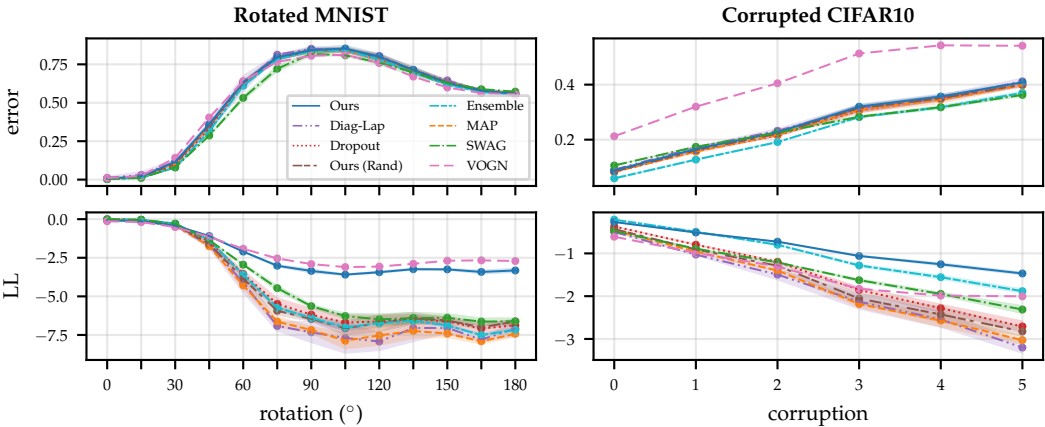

Figure 4: Results on the rotated MNIST (left) and the corrupted CIFAR (right) benchmarks of Ovadia et al. (2019), showing the mean ± std of the error (top) and log-likelihood (bottom) across three different seeds. Subnetwork inference retains better uncertainty calibration and robustness to distribution shift than point estimated networks and other Bayesian deep learning approaches.

approximation for integrals between Gaussians and multi-class cross-entropy likelihoods described in (Gibbs, 1998). We use Wasserstein subnetwork selection to retain only $0.38\%$ of the weights, yielding a subnetwork with only 42,438 weights. This is the largest subnetwork for which we can tractably compute a full covariance matrix. Its size is $42,438^2 \times 4$ Bytes $\approx 7.2$ GB. We use a prior precision of $\lambda = 500$, found via grid search. Finally, to assess to importance of principled subnetwork selection, we also consider the baseline where we select the subnetwork uniformly at random (called *Ours (Rand)*). We perform the following two experiments, with results in Fig. 4. See Appendix B for additional results.

**Rotated MNIST:** Following Ovadia et al. (2019); Antorán et al. (2020), we train all methods on MNIST and evaluate their predictive distributions on increasingly rotated digits. While all methods perform well on the original MNIST test set, their accuracy degrades quickly for rotations larger than 30 degrees. In terms of LL, ensembles perform best out of our baselines. Subnetwork inference obtains significantly larger LL values than almost all baselines, including ensembles. The only exception is VOGN, which achieves slightly better performance. It was also observed in Ovadia et al. (2019) that mean-field variational inference (which VOGN also is an instance of) is very strong on MNIST, but its performance deteriorates on larger datasets. Subnetwork inference makes accurate predictions in-distribution while assigning higher uncertainty than the baselines to out-of-distribution points. **Corrupted CIFAR:** Again following Ovadia et al. (2019); Antorán et al. (2020), we train on CIFAR10 and evaluate on data subject to 16 different corruptions with 5 levels of intensity each (Hendrycks & Dietterich, 2019). Our approach matches a MAP estimated network in terms of predictive error as local linearization makes their predictions the same. Ensembles and SWAG are the most accurate. Even so, subnetwork inference differentiates itself by being the least overconfident, outperforming all baselines in terms of log-likelihood at all corruption levels. Here, VOGN performs rather badly; while this might appear in stark contrast to its strong performance on the MNIST benchmark, the behaviour that mean-field VI performs well on MNIST but not on larger datasets was also observed in Ovadia et al. (2019).

On both benchmarks, we furthermore find that randomly selecting the subnetwork performs substantially worse than using our more principled subnetwork selection strategy. This highlights the importance of the way subnetworks are selected. These results suggest that **subnetwork inference results in better uncertainty calibration and robustness to distribution shift than other popular uncertainty quantification approaches**.

## 6 RELATED WORK

**Bayesian Deep Learning.** There have significant efforts to characterise the posterior distribution over NN weights $p(\boldsymbol{W}|\mathcal{D})$. Hamiltonian Monte Carlo (Neal, 1995) remains the golden standard

for approximate inference in BNNs to this day. Although asymptotically unbiased, sampling based approaches are difficult to scale to the large datasets (Betancourt, 2015). As a result, approaches which find the best surrogate posterior among an approximating family (most often Gaussians) have gained popularity. The first of these was the Laplace approximation, introduced by MacKay (1992), who also proposed approximating the predictive posterior with that of the linearised model (Khan et al., 2019; Immer et al., 2020). The popularisation of larger NN models has made surrogate distributions that capture correlations between weights computationally intractable. Thus, most modern methods make use of the mean field assumption (Blundell et al., 2015; Hernández-Lobato & Adams, 2015; Gal & Ghahramani, 2016; Mishkin et al., 2018; Osawa et al., 2019). This comes at the cost of limited expressivity (Foong et al., 2019a) and empirical under-performance (Ovadia et al., 2019; Antorán et al., 2020) of uncertainty estimates. Our proposed approach recovers predictive posterior expressivity while maintaining tractability by lowering the dimensionality of the weight space considered. This allows us to scale up approximations that *do* consider weight correlations (MacKay, 1992; Louizos & Welling, 2016; Maddox et al., 2019; Ritter et al., 2018).

**Neural Network Linearization.** In the limit of infinite width, NNs converge to Gaussian process (GP) behaviour (Neal, 1995; Matthews, 2017; Garriga-Alonso et al., 2018). Recently, these results have been extended to finite width BNNs when the surrogate posterior is Gaussian (Khan et al., 2019). We draw upon these results to formulate a subnetwork selection strategy for BNNs. Neural linear methods perform inference over only the last layer of a NN, while keeping all other layers fixed (Snoek et al., 2015; Riquelme et al., 2018; Ovadia et al., 2019; Ober & Rasmussen, 2019; Pinsler et al., 2019; Kristiadi et al., 2020). These represent a different generalised linear model in which the basis functions are defined by the $l-1$ first layers of a NN. They can also be viewed as a special case of subnetwork inference, in which the subnetwork is simply defined to be the last NN layer.

**Inference over Subspaces.** The subfield of NN pruning aims to increase the computational efficiency of NNs by identifying the smallest subset of weights which are required to make accurate predictions. Approaches trade-off computational cost with compression efficiency, ranging from those that require multiple training runs (Frankle & Carbin, 2019) to those that prune before training (Wang et al., 2020). Our work differs in that it retains all NN weights but aims to find a small subset over which to perform probabilistic reasoning. More closely related work to ours is that of Izmailov et al. (2019), who propose to perform inference over a low-dimensional subspace of weights; e.g. one constructed from the principal components of the SGD trajectory. Moreover, several recent approaches use low-rank parameterizations of approximate posteriors in the context of variational inference (Rossi et al., 2019; Swiatkowski et al., 2020; Dusenberry et al., 2020). This could also be viewed as doing inference over an implicit subspace of weight space. In contrast, we propose a technique to find subsets of weights which are relevant to predictive uncertainty, i.e., we identify axis aligned subspaces. Finally, there have been recent works studying neural network sparsity / pruning from a Bayesian perspective (Ghosh & Doshi-Velez, 2017; Polson & Ročková, 2018; Cui et al., 2020; Louizos et al., 2017; Molchanov et al., 2017; Gomez et al., 2019; Lee et al., 2018). While these seem conceptually related at first glance, their goal is fundamentally different to ours: While those methods aim to perform model selection / sparsification by either explicitly or implicitly pruning unnecessary weights, our goal is to make inference more tractable. More precisely, while those sparse Bayesian deep learning methods prune individual *weights*, we instead just prune the *variances* of certain weights, which, importantly retains the full predictive power of the full network to retain high predictive accuracy.

## 7 CONCLUSION

In this paper, we develop a *practical* and *scalable* method for expressive yet tractable probabilistic inference in deep neural networks. We approximate the posterior over a subset of the weights while keeping all other weights deterministic. Computational cost is decoupled from network size, allowing us to *scale* expressive approximations, such as full-covariance Gaussian distributions, to real-world sized NNs. Our approach can be applied post-hoc to any pre-trained model, making it particularly attractive for practical use. Our empirical analysis suggests that subnetwork inference 1) is more expressive and retains more uncertainty than crude approximations over the full network, 2) allows us to employ larger NNs, which fit a broader range of functions, without sacrificing the quality of our uncertainty estimates, and 3) is competitive with state-of-the-art uncertainty quantification methods, like deep ensembles (Lakshminarayanan et al., 2017), on real-world scale problems.

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

## A  PROOFS FOR THE THEORETICAL RESULTS

We now provide the proofs for the results in Section 4.

### A.1  PROOF OF PROPOSITION 1

*Proof.* Note that the posterior distributions $\widetilde{p}(\mathbf{w}|\boldsymbol{y}, \boldsymbol{\Phi})$ and $p_S(\mathbf{w}|\boldsymbol{y}, \boldsymbol{\Phi})$ are both Gaussian. We thus consider the squared 2-Wasserstein distance between two Gaussian distributions $\mathcal{N}(\boldsymbol{\mu}_1, \boldsymbol{\Sigma}_1)$ and $\mathcal{N}(\boldsymbol{\mu}_2, \boldsymbol{\Sigma}_2)$, which has the following closed-form expression (Givens et al., 1984)[4]:

$$W\left[\mathcal{N}\left(\boldsymbol{\mu}_1, \boldsymbol{\Sigma}_1\right) \| \mathcal{N}\left(\boldsymbol{\mu}_2, \boldsymbol{\Sigma}_2\right)\right] = \|\boldsymbol{\mu}_1 - \boldsymbol{\mu}_2\|_2^2 + \text{trace}\left(\boldsymbol{\Sigma}_1 + \boldsymbol{\Sigma}_2 - 2\left(\boldsymbol{\Sigma}_1\boldsymbol{\Sigma}_2\right)^{1/2}\right) . \tag{13}$$

Plugging in $\boldsymbol{\mu}_1 = \boldsymbol{\mu}_2 = \mathbf{w}_{MAP}$, $\boldsymbol{\Sigma}_1 = H^{-1}$ and $\boldsymbol{\Sigma}_2 = \boldsymbol{M}_S \odot H^{-1}$, we obtain

$$\begin{aligned}
W &\left[\widetilde{p}(\mathbf{w}|\boldsymbol{y}, \boldsymbol{X}) \| p_S(\mathbf{w}|\boldsymbol{y}, \boldsymbol{X})\right] \\
&= W\left[\mathcal{N}(\mathbf{w}_{MAP}, H^{-1}) \| \mathcal{N}(\mathbf{w}_{MAP}, \boldsymbol{M}_S \odot H^{-1})\right] \\
&= \|\mathbf{w}_{MAP} - \mathbf{w}_{MAP}\|_2^2 + \text{trace}\left(H^{-1} + (\boldsymbol{M}_S \odot H^{-1}) - 2\left(H^{-1}(\boldsymbol{M}_S \odot H^{-1})\right)^{1/2}\right) \\
&= \text{trace}\left((\mathbf{1} + \boldsymbol{M}_S) \odot H^{-1}\right) - \text{trace}\left(2\left(H^{-1}\left(\boldsymbol{M}_S \odot H^{-1}\right)\right)^{1/2}\right) \\
&= \sum_{d=1}^{D} (1 + m_{dd})\,\sigma_d^2 - \text{trace}\left(2\left(H^{-1}\left(\boldsymbol{M}_S \odot H^{-1}\right)\right)^{1/2}\right) \qquad \square
\end{aligned}$$

### A.2  PROOF OF COROLLARY 1.1

*Proof.* For $H^{-1} = \text{diag}(\sigma_1^2, \ldots, \sigma_D^2)$, the Wasserstein posterior gap in Eq. (12) simplifies to

$$W\left[\widetilde{p}(\mathbf{w}|\boldsymbol{y}, \boldsymbol{\Phi}) \| p_S(\mathbf{w}|\boldsymbol{y}, \boldsymbol{\Phi})\right] = \sum_{d=1}^{D}\left((1 + m_{dd})\,\sigma_d^2 - 2m_{dd}\sigma_d^2\right) . \tag{14}$$

The optimal subnetwork selection strategy amounts to choosing the binary vector $\boldsymbol{m} = [m_{dd}]_{d=1}^{D}$ with $\sum_{d=1}^{D} m_d = S$ (i.e., we select $S$ out of $D$ parameters) s.t. the posterior gap in Eq. (14) is *minimized*. Observing that the contribution of the $d$-th parameter to the posterior gap is $(1+1)\sigma_d^2 - 1 \times 2\sigma_d^2 = 0$ if it is selected (i.e. if $m_{dd} = 1$), and $(1+0)\sigma_d^2 - 0 \times 2\sigma_d^2 = \sigma_d^2$ if it is *not* selected (i.e. if $m_{dd} = 0$), we see that the optimal subnetwork comprises of the $S$ weights with the *largest* variances $\sigma_d^2$. $\square$

## B  ADDITIONAL IMAGE CLASSIFICATION RESULTS

Table 1: AUC-ROC scores for out-of-distribution detection, using CIFAR10 vs SVHN and MNIST vs FashionMNIST as in- (source) and out-of-distribution (target) datasets, respectively (Nalisnick et al., 2019).

| SOURCE | TARGET | OURS | OURS (RAND) | DROPOUT | DIAG-LAP | ENSEMBLE | MAP | SWAG |
|---|---|---|---|---|---|---|---|---|
| CIFAR10 | SVHN | 0.85±0.03 | 0.86±0.02 | 0.85±0.01 | 0.86±0.02 | 0.91±0.00 | 0.86±0.02 | 0.83±0.00 |
| MNIST | Fashion | 0.92±0.05 | 0.75±0.02 | 0.82±0.12 | 0.75±0.01 | 0.90±0.09 | 0.72±0.03 | 0.97±0.01 |

Table 2: MNIST – no rotation.

| | OURS | OURS (RAND) | DROPOUT | DIAG-LAP | ENSEMBLE | MAP | SWAG | VOGN |
|---|---|---|---|---|---|---|---|---|
| LL | −0.07±0.01 | −0.01±0.00 | −0.01±0.00 | −0.04±0.03 | −0.01±0.00 | −0.01±0.00 | −0.01±0.00 | −0.14±nan |
| error | 0.01±0.00 | 0.00±0.00 | 0.00±0.00 | 0.01±0.01 | 0.00±0.00 | 0.00±0.00 | 0.00±0.00 | 0.01±nan |
| ECE | 0.05±0.01 | 0.00±0.00 | 0.00±0.00 | 0.00±0.00 | 0.00±0.00 | 0.00±0.00 | 0.00±0.00 | 0.10±nan |
| brier score | 0.02±0.00 | 0.01±0.00 | 0.01±0.00 | 0.02±0.01 | 0.01±0.00 | 0.01±0.00 | 0.01±0.00 | 0.04±nan |

---

[4]This also holds for our case of a degenerate Gaussian with singular covariance matrix (Givens et al., 1984).

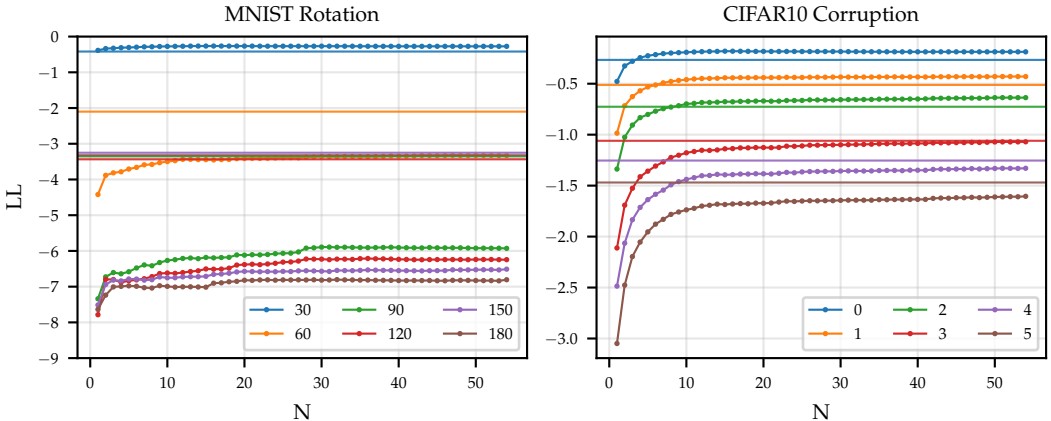

Figure 5: Rotated MNIST (left) and Corrupted CIFAR10 (right) results for deep ensembles (Lakshminarayanan et al., 2017) with large numbers of ensemble members (i.e. up to 55). Horizontal axis denotes number of ensemble members, and vertical axis denotes performance in terms of log-likelihood. Straight horizontal lines correspond to the performance of our method, as a reference. Colors denote different levels of rotation (left) and corruption (right). It can clearly be observed that the performance of deep ensembles saturates after around 15 ensemble members, meaning that adding more members yields strongly diminishing returns. This is in agreement with recent works (Antorán et al., 2020; Ashukha et al., 2020a; Lobacheva et al., 2020). Our method significantly outperforms even very large deep ensembles, especially for high degrees of rotation/corruption.

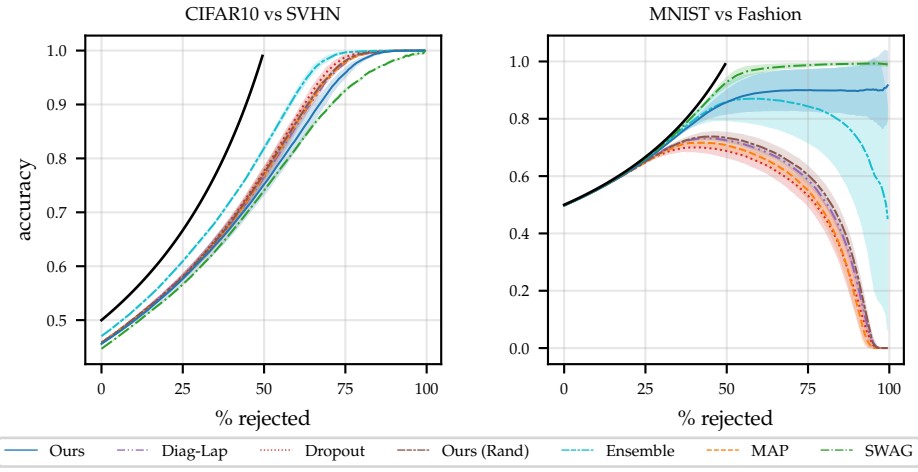

Figure 6: Rejection-classification plots. We simulate a realistic OOD rejection scenario (Filos et al., 2019b) by jointly evaluating our models on an in-distribution and an OOD test set. We allow our methods to reject increasing proportions of the data based on predictive entropy before classifying the rest. All predictions on OOD samples are treated as incorrect. Following (Nalisnick et al., 2019), we use CIFAR10 vs SVHN and MNIST vs FashionMNIST as in- and out-of-distribution datasets, respectively. Note that the SVHN test set is randomly sub-sampled down to a size of 10,000.

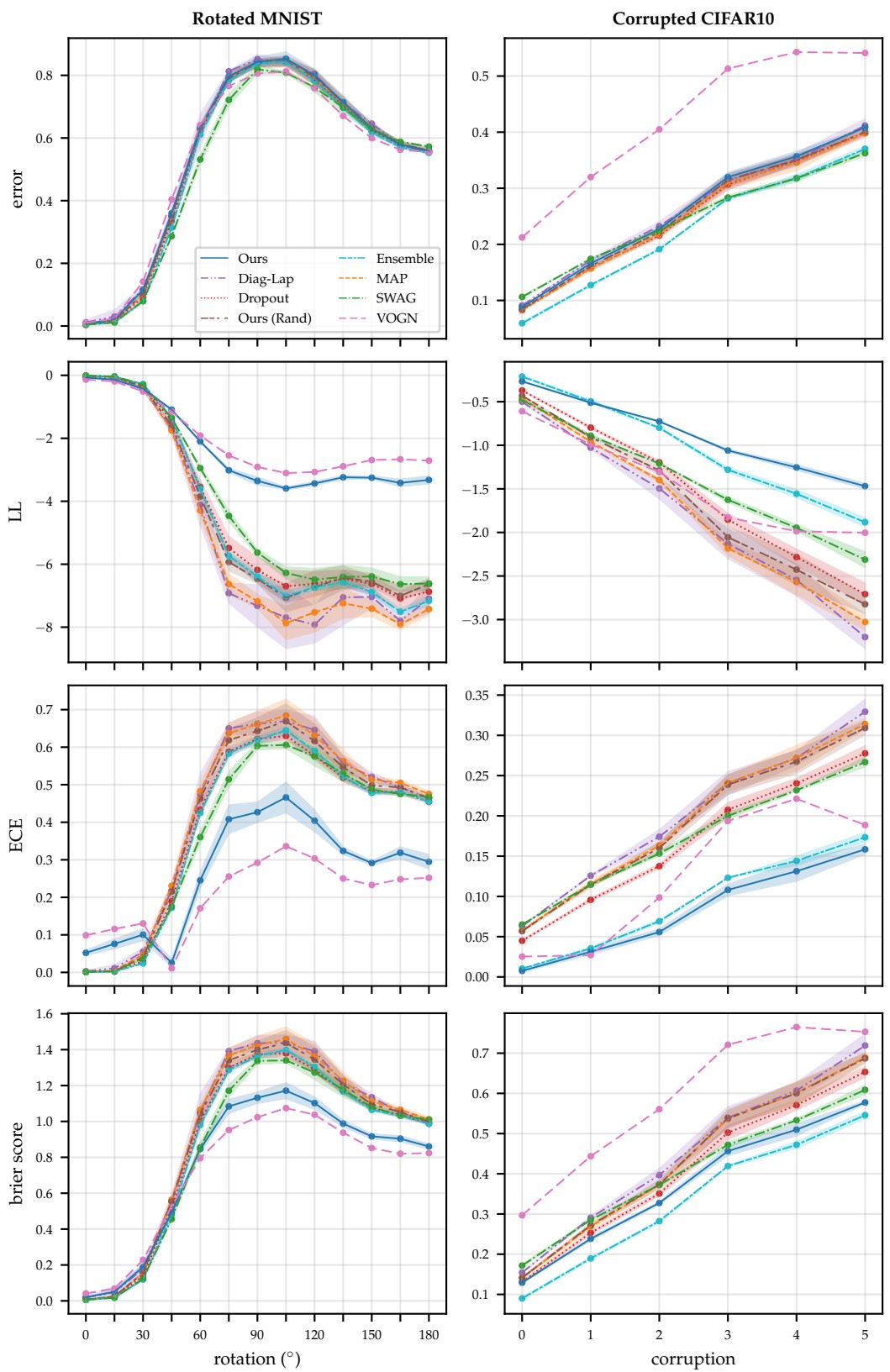

Figure 7: Full MNIST rotation and CIFAR10 corruption results, for ResNet-18, reporting predictive error, log-likelihood (LL), expected calibration error (ECE) and brier score, respectively (from top to bottom).

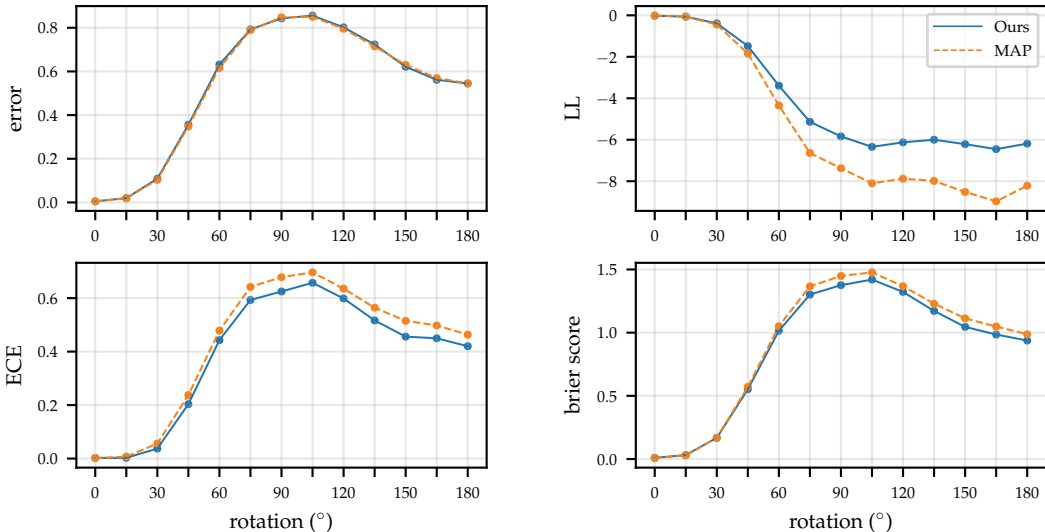

Figure 8: MNIST rotation results for ResNet-50, reporting predictive error, log-likelihood (LL), expected calibration error (ECE) and brier score. We choose a subnetwork containing only 0.167% (39,190 / 23,466,560) of the parameters of the full network. We see that subnetwork inference still results in an improvement in the calibration of predictive uncertainty. As expected, however, for ResNet-50 the improvement over MAP is smaller than for ResNet-18 where we were able to choose a subnetwork containing 0.38% of the parameters.

Table 3: MNIST – 15° rotation.

| | OURS | OURS (RAND) | DROPOUT | DIAG-LAP | ENSEMBLE | MAP | SWAG | VOGN |
|---|---|---|---|---|---|---|---|---|
| LL | $-0.14_{\pm 0.02}$ | $-0.05_{\pm 0.00}$ | $-0.05_{\pm 0.00}$ | $-0.11_{\pm 0.08}$ | $-0.04_{\pm 0.00}$ | $-0.05_{\pm 0.00}$ | $-0.04_{\pm 0.00}$ | $-0.19_{\pm nan}$ |
| error | $0.02_{\pm 0.00}$ | $0.02_{\pm 0.00}$ | $0.01_{\pm 0.00}$ | $0.03_{\pm 0.02}$ | $0.01_{\pm 0.00}$ | $0.02_{\pm 0.00}$ | $0.01_{\pm 0.00}$ | $0.02_{\pm nan}$ |
| ECE | $0.08_{\pm 0.01}$ | $0.00_{\pm 0.00}$ | $0.00_{\pm 0.00}$ | $0.01_{\pm 0.01}$ | $0.00_{\pm 0.00}$ | $0.00_{\pm 0.00}$ | $0.00_{\pm 0.00}$ | $0.12_{\pm nan}$ |
| brier score | $0.05_{\pm 0.01}$ | $0.03_{\pm 0.00}$ | $0.02_{\pm 0.00}$ | $0.05_{\pm 0.03}$ | $0.02_{\pm 0.00}$ | $0.02_{\pm 0.00}$ | $0.02_{\pm 0.00}$ | $0.07_{\pm nan}$ |

Table 4: MNIST – 30° rotation.

| | OURS | OURS (RAND) | DROPOUT | DIAG-LAP | ENSEMBLE | MAP | SWAG | VOGN |
|---|---|---|---|---|---|---|---|---|
| LL | $-0.42_{\pm 0.04}$ | $-0.36_{\pm 0.01}$ | $-0.32_{\pm 0.02}$ | $-0.44_{\pm 0.06}$ | $-0.28_{\pm 0.02}$ | $-0.39_{\pm 0.01}$ | $-0.30_{\pm 0.00}$ | $-0.51_{\pm nan}$ |
| error | $0.11_{\pm 0.01}$ | $0.10_{\pm 0.00}$ | $0.09_{\pm 0.01}$ | $0.12_{\pm 0.01}$ | $0.08_{\pm 0.01}$ | $0.10_{\pm 0.00}$ | $0.08_{\pm 0.00}$ | $0.14_{\pm nan}$ |
| ECE | $0.10_{\pm 0.02}$ | $0.04_{\pm 0.01}$ | $0.03_{\pm 0.00}$ | $0.06_{\pm 0.01}$ | $0.02_{\pm 0.00}$ | $0.05_{\pm 0.00}$ | $0.04_{\pm 0.00}$ | $0.13_{\pm nan}$ |
| brier score | $0.19_{\pm 0.02}$ | $0.16_{\pm 0.00}$ | $0.14_{\pm 0.01}$ | $0.18_{\pm 0.02}$ | $0.12_{\pm 0.01}$ | $0.16_{\pm 0.00}$ | $0.12_{\pm 0.00}$ | $0.23_{\pm nan}$ |

Table 5: MNIST – 45° rotation.

| | OURS | OURS (RAND) | DROPOUT | DIAG-LAP | ENSEMBLE | MAP | SWAG | VOGN |
|---|---|---|---|---|---|---|---|---|
| midrule LL | $-1.09_{\pm 0.03}$ | $-1.60_{\pm 0.05}$ | $-1.44_{\pm 0.11}$ | $-1.68_{\pm 0.20}$ | $-1.36_{\pm 0.07}$ | $-1.75_{\pm 0.06}$ | $-1.35_{\pm 0.02}$ | $-1.15_{\pm nan}$ |
| error | $0.36_{\pm 0.01}$ | $0.35_{\pm 0.01}$ | $0.33_{\pm 0.01}$ | $0.35_{\pm 0.03}$ | $0.31_{\pm 0.01}$ | $0.35_{\pm 0.01}$ | $0.29_{\pm 0.00}$ | $0.40_{\pm nan}$ |
| ECE | $0.03_{\pm 0.01}$ | $0.22_{\pm 0.01}$ | $0.19_{\pm 0.02}$ | $0.22_{\pm 0.02}$ | $0.17_{\pm 0.01}$ | $0.23_{\pm 0.01}$ | $0.18_{\pm 0.00}$ | $0.01_{\pm nan}$ |
| brier score | $0.49_{\pm 0.02}$ | $0.55_{\pm 0.02}$ | $0.52_{\pm 0.02}$ | $0.55_{\pm 0.04}$ | $0.48_{\pm 0.02}$ | $0.56_{\pm 0.02}$ | $0.46_{\pm 0.01}$ | $0.53_{\pm nan}$ |

Table 6: MNIST – 60° rotation.

| | OURS | OURS (RAND) | DROPOUT | DIAG-LAP | ENSEMBLE | MAP | SWAG | VOGN |
|---|---|---|---|---|---|---|---|---|
| LL | $-2.10_{\pm 0.03}$ | $-3.85_{\pm 0.18}$ | $-3.54_{\pm 0.23}$ | $-4.11_{\pm 0.66}$ | $-3.60_{\pm 0.10}$ | $-4.29_{\pm 0.21}$ | $-2.95_{\pm 0.08}$ | $-1.92_{\pm nan}$ |
| error | $0.63_{\pm 0.01}$ | $0.63_{\pm 0.01}$ | $0.62_{\pm 0.01}$ | $0.62_{\pm 0.05}$ | $0.61_{\pm 0.01}$ | $0.63_{\pm 0.01}$ | $0.53_{\pm 0.02}$ | $0.64_{\pm nan}$ |
| ECE | $0.25_{\pm 0.02}$ | $0.46_{\pm 0.02}$ | $0.43_{\pm 0.02}$ | $0.47_{\pm 0.06}$ | $0.42_{\pm 0.01}$ | $0.48_{\pm 0.02}$ | $0.36_{\pm 0.02}$ | $0.17_{\pm nan}$ |
| brier score | $0.85_{\pm 0.02}$ | $1.04_{\pm 0.03}$ | $1.00_{\pm 0.03}$ | $1.05_{\pm 0.10}$ | $0.98_{\pm 0.02}$ | $1.07_{\pm 0.03}$ | $0.86_{\pm 0.03}$ | $0.80_{\pm nan}$ |

Table 7: MNIST – $75°$ rotation.

|  | OURS | OURS (RAND) | DROPOUT | DIAG-LAP | ENSEMBLE | MAP | SWAG | VOGN |
|---|---|---|---|---|---|---|---|---|
| LL | $-3.02_{\pm 0.07}$ | $-5.93_{\pm 0.28}$ | $-5.49_{\pm 0.38}$ | $-6.92_{\pm 0.32}$ | $-5.74_{\pm 0.15}$ | $-6.63_{\pm 0.33}$ | $-4.46_{\pm 0.18}$ | $-2.54_{\pm nan}$ |
| error | $0.80_{\pm 0.02}$ | $0.79_{\pm 0.01}$ | $0.79_{\pm 0.01}$ | $0.81_{\pm 0.00}$ | $0.78_{\pm 0.01}$ | $0.79_{\pm 0.01}$ | $0.72_{\pm 0.02}$ | $0.77_{\pm nan}$ |
| ECE | $0.41_{\pm 0.04}$ | $0.62_{\pm 0.03}$ | $0.59_{\pm 0.01}$ | $0.65_{\pm 0.01}$ | $0.58_{\pm 0.01}$ | $0.64_{\pm 0.03}$ | $0.51_{\pm 0.02}$ | $0.26_{\pm nan}$ |
| brier score | $1.08_{\pm 0.04}$ | $1.34_{\pm 0.04}$ | $1.30_{\pm 0.02}$ | $1.39_{\pm 0.01}$ | $1.29_{\pm 0.02}$ | $1.37_{\pm 0.04}$ | $1.17_{\pm 0.04}$ | $0.95_{\pm nan}$ |

Table 8: MNIST – $90°$ rotation.

|  | OURS | OURS (RAND) | DROPOUT | DIAG-LAP | ENSEMBLE | MAP | SWAG | VOGN |
|---|---|---|---|---|---|---|---|---|
| LL | $-3.35_{\pm 0.13}$ | $-6.46_{\pm 0.15}$ | $-6.18_{\pm 0.41}$ | $-7.32_{\pm 0.67}$ | $-6.39_{\pm 0.17}$ | $-7.18_{\pm 0.22}$ | $-5.63_{\pm 0.12}$ | $-2.91_{\pm nan}$ |
| error | $0.84_{\pm 0.02}$ | $0.84_{\pm 0.01}$ | $0.84_{\pm 0.01}$ | $0.85_{\pm 0.01}$ | $0.84_{\pm 0.01}$ | $0.84_{\pm 0.01}$ | $0.82_{\pm 0.01}$ | $0.81_{\pm nan}$ |
| ECE | $0.43_{\pm 0.03}$ | $0.64_{\pm 0.04}$ | $0.62_{\pm 0.01}$ | $0.66_{\pm 0.03}$ | $0.62_{\pm 0.01}$ | $0.66_{\pm 0.04}$ | $0.60_{\pm 0.01}$ | $0.29_{\pm nan}$ |
| brier score | $1.13_{\pm 0.03}$ | $1.40_{\pm 0.05}$ | $1.37_{\pm 0.01}$ | $1.44_{\pm 0.04}$ | $1.36_{\pm 0.01}$ | $1.43_{\pm 0.05}$ | $1.34_{\pm 0.02}$ | $1.02_{\pm nan}$ |

Table 9: MNIST – $105°$ rotation.

|  | OURS | OURS (RAND) | DROPOUT | DIAG-LAP | ENSEMBLE | MAP | SWAG | VOGN |
|---|---|---|---|---|---|---|---|---|
| LL | $-3.59_{\pm 0.05}$ | $-7.06_{\pm 0.45}$ | $-6.70_{\pm 0.52}$ | $-7.69_{\pm 0.99}$ | $-7.01_{\pm 0.17}$ | $-7.87_{\pm 0.53}$ | $-6.28_{\pm 0.19}$ | $-3.10_{\pm nan}$ |
| error | $0.85_{\pm 0.02}$ | $0.84_{\pm 0.02}$ | $0.84_{\pm 0.01}$ | $0.85_{\pm 0.01}$ | $0.84_{\pm 0.01}$ | $0.84_{\pm 0.02}$ | $0.81_{\pm 0.00}$ | $0.81_{\pm nan}$ |
| ECE | $0.47_{\pm 0.04}$ | $0.67_{\pm 0.05}$ | $0.63_{\pm 0.01}$ | $0.67_{\pm 0.03}$ | $0.64_{\pm 0.01}$ | $0.68_{\pm 0.04}$ | $0.61_{\pm 0.01}$ | $0.34_{\pm nan}$ |
| brier score | $1.17_{\pm 0.05}$ | $1.44_{\pm 0.07}$ | $1.38_{\pm 0.02}$ | $1.44_{\pm 0.04}$ | $1.40_{\pm 0.01}$ | $1.46_{\pm 0.07}$ | $1.34_{\pm 0.02}$ | $1.07_{\pm nan}$ |

Table 10: MNIST – $120°$ rotation.

|  | OURS | OURS (RAND) | DROPOUT | DIAG-LAP | ENSEMBLE | MAP | SWAG | VOGN |
|---|---|---|---|---|---|---|---|---|
| LL | $-3.43_{\pm 0.07}$ | $-6.73_{\pm 0.53}$ | $-6.62_{\pm 0.39}$ | $-7.92_{\pm 0.59}$ | $-6.73_{\pm 0.11}$ | $-7.53_{\pm 0.63}$ | $-6.49_{\pm 0.36}$ | $-3.07_{\pm nan}$ |
| error | $0.80_{\pm 0.02}$ | $0.79_{\pm 0.02}$ | $0.78_{\pm 0.01}$ | $0.81_{\pm 0.01}$ | $0.78_{\pm 0.01}$ | $0.79_{\pm 0.02}$ | $0.76_{\pm 0.02}$ | $0.76_{\pm nan}$ |
| ECE | $0.40_{\pm 0.03}$ | $0.62_{\pm 0.05}$ | $0.58_{\pm 0.01}$ | $0.65_{\pm 0.04}$ | $0.59_{\pm 0.01}$ | $0.63_{\pm 0.04}$ | $0.58_{\pm 0.03}$ | $0.30_{\pm nan}$ |
| brier score | $1.10_{\pm 0.03}$ | $1.35_{\pm 0.07}$ | $1.29_{\pm 0.02}$ | $1.39_{\pm 0.06}$ | $1.30_{\pm 0.01}$ | $1.36_{\pm 0.07}$ | $1.27_{\pm 0.04}$ | $1.04_{\pm nan}$ |

Table 11: MNIST – $135°$ rotation.

|  | OURS | OURS (RAND) | DROPOUT | DIAG-LAP | ENSEMBLE | MAP | SWAG | VOGN |
|---|---|---|---|---|---|---|---|---|
| LL | $-3.24_{\pm 0.06}$ | $-6.43_{\pm 0.38}$ | $-6.46_{\pm 0.28}$ | $-7.05_{\pm 0.88}$ | $-6.57_{\pm 0.10}$ | $-7.24_{\pm 0.48}$ | $-6.40_{\pm 0.37}$ | $-2.89_{\pm nan}$ |
| error | $0.71_{\pm 0.02}$ | $0.71_{\pm 0.02}$ | $0.70_{\pm 0.01}$ | $0.71_{\pm 0.01}$ | $0.70_{\pm 0.01}$ | $0.71_{\pm 0.02}$ | $0.70_{\pm 0.02}$ | $0.67_{\pm nan}$ |
| ECE | $0.32_{\pm 0.01}$ | $0.55_{\pm 0.03}$ | $0.52_{\pm 0.01}$ | $0.56_{\pm 0.02}$ | $0.52_{\pm 0.01}$ | $0.56_{\pm 0.03}$ | $0.53_{\pm 0.02}$ | $0.25_{\pm nan}$ |
| brier score | $0.99_{\pm 0.02}$ | $1.21_{\pm 0.05}$ | $1.17_{\pm 0.02}$ | $1.22_{\pm 0.04}$ | $1.17_{\pm 0.01}$ | $1.23_{\pm 0.05}$ | $1.18_{\pm 0.04}$ | $0.94_{\pm nan}$ |

Table 12: MNIST – $150°$ rotation.

|  | OURS | OURS (RAND) | DROPOUT | DIAG-LAP | ENSEMBLE | MAP | SWAG | VOGN |
|---|---|---|---|---|---|---|---|---|
| LL | $-3.25_{\pm 0.05}$ | $-6.56_{\pm 0.18}$ | $-6.62_{\pm 0.33}$ | $-7.04_{\pm 0.36}$ | $-6.88_{\pm 0.11}$ | $-7.41_{\pm 0.25}$ | $-6.39_{\pm 0.27}$ | $-2.69_{\pm nan}$ |
| error | $0.63_{\pm 0.02}$ | $0.63_{\pm 0.01}$ | $0.63_{\pm 0.00}$ | $0.65_{\pm 0.01}$ | $0.62_{\pm 0.01}$ | $0.63_{\pm 0.01}$ | $0.63_{\pm 0.01}$ | $0.60_{\pm nan}$ |
| ECE | $0.29_{\pm 0.01}$ | $0.50_{\pm 0.01}$ | $0.48_{\pm 0.01}$ | $0.52_{\pm 0.01}$ | $0.48_{\pm 0.01}$ | $0.51_{\pm 0.01}$ | $0.49_{\pm 0.01}$ | $0.23_{\pm nan}$ |
| brier score | $0.92_{\pm 0.02}$ | $1.10_{\pm 0.02}$ | $1.07_{\pm 0.01}$ | $1.13_{\pm 0.02}$ | $1.06_{\pm 0.01}$ | $1.11_{\pm 0.02}$ | $1.08_{\pm 0.02}$ | $0.85_{\pm nan}$ |

Table 13: MNIST – $165°$ rotation.

|  | OURS | OURS (RAND) | DROPOUT | DIAG-LAP | ENSEMBLE | MAP | SWAG | VOGN |
|---|---|---|---|---|---|---|---|---|
| LL | $-3.42_{\pm 0.12}$ | $-7.01_{\pm 0.15}$ | $-7.08_{\pm 0.39}$ | $-7.80_{\pm 0.12}$ | $-7.51_{\pm 0.11}$ | $-7.91_{\pm 0.18}$ | $-6.63_{\pm 0.24}$ | $-2.67_{\pm nan}$ |
| error | $0.58_{\pm 0.01}$ | $0.58_{\pm 0.01}$ | $0.58_{\pm 0.01}$ | $0.58_{\pm 0.00}$ | $0.57_{\pm 0.01}$ | $0.58_{\pm 0.01}$ | $0.59_{\pm 0.00}$ | $0.56_{\pm nan}$ |
| ECE | $0.32_{\pm 0.02}$ | $0.49_{\pm 0.01}$ | $0.48_{\pm 0.01}$ | $0.49_{\pm 0.01}$ | $0.48_{\pm 0.00}$ | $0.51_{\pm 0.01}$ | $0.48_{\pm 0.00}$ | $0.25_{\pm nan}$ |
| brier score | $0.90_{\pm 0.02}$ | $1.05_{\pm 0.01}$ | $1.04_{\pm 0.01}$ | $1.05_{\pm 0.01}$ | $1.03_{\pm 0.01}$ | $1.07_{\pm 0.02}$ | $1.03_{\pm 0.01}$ | $0.82_{\pm nan}$ |

Table 14: MNIST – $180°$ rotation.

|  | OURS | OURS (RAND) | DROPOUT | DIAG-LAP | ENSEMBLE | MAP | SWAG | VOGN |
|---|---|---|---|---|---|---|---|---|
| LL | $-3.32_{\pm 0.13}$ | $-6.63_{\pm 0.18}$ | $-6.87_{\pm 0.32}$ | $-7.10_{\pm 0.47}$ | $-7.16_{\pm 0.16}$ | $-7.43_{\pm 0.20}$ | $-6.61_{\pm 0.22}$ | $-2.71_{\pm nan}$ |
| error | $0.56_{\pm 0.01}$ | $0.56_{\pm 0.01}$ | $0.56_{\pm 0.00}$ | $0.55_{\pm 0.01}$ | $0.55_{\pm 0.00}$ | $0.56_{\pm 0.01}$ | $0.57_{\pm 0.00}$ | $0.55_{\pm nan}$ |
| ECE | $0.29_{\pm 0.02}$ | $0.46_{\pm 0.01}$ | $0.45_{\pm 0.00}$ | $0.46_{\pm 0.00}$ | $0.46_{\pm 0.01}$ | $0.48_{\pm 0.01}$ | $0.47_{\pm 0.01}$ | $0.25_{\pm nan}$ |
| brier score | $0.86_{\pm 0.02}$ | $1.00_{\pm 0.01}$ | $0.99_{\pm 0.01}$ | $0.99_{\pm 0.01}$ | $0.99_{\pm 0.00}$ | $1.01_{\pm 0.02}$ | $1.01_{\pm 0.01}$ | $0.82_{\pm nan}$ |

Table 15: CIFAR10 – no corruption.

|  | OURS | OURS (RAND) | DROPOUT | DIAG-LAP | ENSEMBLE | MAP | SWAG | VOGN |
|---|---|---|---|---|---|---|---|---|
| LL | $-0.27_{\pm 0.00}$ | $-0.43_{\pm 0.01}$ | $-0.37_{\pm 0.01}$ | $-0.50_{\pm 0.02}$ | $-0.21_{\pm 0.01}$ | $-0.46_{\pm 0.02}$ | $-0.48_{\pm 0.01}$ | $-0.61_{\pm nan}$ |
| error | $0.09_{\pm 0.00}$ | $0.08_{\pm 0.00}$ | $0.08_{\pm 0.00}$ | $0.09_{\pm 0.00}$ | $0.06_{\pm 0.00}$ | $0.08_{\pm 0.00}$ | $0.11_{\pm 0.00}$ | $0.21_{\pm nan}$ |
| ECE | $0.01_{\pm 0.00}$ | $0.06_{\pm 0.00}$ | $0.04_{\pm 0.00}$ | $0.06_{\pm 0.00}$ | $0.01_{\pm 0.00}$ | $0.06_{\pm 0.00}$ | $0.07_{\pm 0.00}$ | $0.03_{\pm nan}$ |
| brier score | $0.13_{\pm 0.00}$ | $0.14_{\pm 0.00}$ | $0.13_{\pm 0.00}$ | $0.15_{\pm 0.00}$ | $0.09_{\pm 0.00}$ | $0.14_{\pm 0.00}$ | $0.17_{\pm 0.00}$ | $0.30_{\pm nan}$ |

Table 16: CIFAR10 – level 1 corruption.

|  | OURS | OURS (RAND) | DROPOUT | DIAG-LAP | ENSEMBLE | MAP | SWAG | VOGN |
|---|---|---|---|---|---|---|---|---|
| LL | $-0.51_{\pm 0.01}$ | $-0.91_{\pm 0.01}$ | $-0.80_{\pm 0.02}$ | $-1.03_{\pm 0.02}$ | $-0.50_{\pm 0.02}$ | $-0.96_{\pm 0.02}$ | $-0.89_{\pm 0.02}$ | $-0.99_{\pm nan}$ |
| error | $0.17_{\pm 0.01}$ | $0.16_{\pm 0.00}$ | $0.16_{\pm 0.00}$ | $0.17_{\pm 0.00}$ | $0.13_{\pm 0.00}$ | $0.16_{\pm 0.00}$ | $0.17_{\pm 0.00}$ | $0.32_{\pm nan}$ |
| ECE | $0.03_{\pm 0.00}$ | $0.11_{\pm 0.00}$ | $0.10_{\pm 0.00}$ | $0.13_{\pm 0.00}$ | $0.04_{\pm 0.00}$ | $0.12_{\pm 0.01}$ | $0.11_{\pm 0.00}$ | $0.03_{\pm nan}$ |
| brier score | $0.24_{\pm 0.00}$ | $0.27_{\pm 0.00}$ | $0.25_{\pm 0.00}$ | $0.29_{\pm 0.00}$ | $0.19_{\pm 0.00}$ | $0.27_{\pm 0.01}$ | $0.29_{\pm 0.00}$ | $0.44_{\pm nan}$ |

Table 17: CIFAR10 – level 2 corruption.

|  | OURS | OURS (RAND) | DROPOUT | DIAG-LAP | ENSEMBLE | MAP | SWAG | VOGN |
|---|---|---|---|---|---|---|---|---|
| LL | $-0.73_{\pm 0.01}$ | $-1.29_{\pm 0.06}$ | $-1.20_{\pm 0.02}$ | $-1.50_{\pm 0.12}$ | $-0.80_{\pm 0.01}$ | $-1.40_{\pm 0.03}$ | $-1.21_{\pm 0.00}$ | $-1.31_{\pm nan}$ |
| error | $0.23_{\pm 0.00}$ | $0.22_{\pm 0.01}$ | $0.22_{\pm 0.00}$ | $0.23_{\pm 0.01}$ | $0.19_{\pm 0.00}$ | $0.22_{\pm 0.00}$ | $0.22_{\pm 0.00}$ | $0.40_{\pm nan}$ |
| ECE | $0.06_{\pm 0.00}$ | $0.16_{\pm 0.01}$ | $0.14_{\pm 0.00}$ | $0.17_{\pm 0.01}$ | $0.07_{\pm 0.00}$ | $0.16_{\pm 0.00}$ | $0.15_{\pm 0.00}$ | $0.10_{\pm nan}$ |
| brier score | $0.33_{\pm 0.00}$ | $0.37_{\pm 0.01}$ | $0.35_{\pm 0.01}$ | $0.40_{\pm 0.02}$ | $0.28_{\pm 0.00}$ | $0.37_{\pm 0.01}$ | $0.37_{\pm 0.00}$ | $0.56_{\pm nan}$ |

Table 18: CIFAR10 – level 3 corruption.

|  | OURS | OURS (RAND) | DROPOUT | DIAG-LAP | ENSEMBLE | MAP | SWAG | VOGN |
|---|---|---|---|---|---|---|---|---|
| LL | $-1.06_{\pm 0.02}$ | $-2.06_{\pm 0.12}$ | $-1.85_{\pm 0.07}$ | $-2.13_{\pm 0.17}$ | $-1.28_{\pm 0.03}$ | $-2.18_{\pm 0.08}$ | $-1.63_{\pm 0.03}$ | $-1.83_{\pm nan}$ |
| error | $0.32_{\pm 0.01}$ | $0.31_{\pm 0.01}$ | $0.31_{\pm 0.01}$ | $0.31_{\pm 0.01}$ | $0.28_{\pm 0.00}$ | $0.31_{\pm 0.01}$ | $0.28_{\pm 0.00}$ | $0.51_{\pm nan}$ |
| ECE | $0.11_{\pm 0.01}$ | $0.24_{\pm 0.01}$ | $0.21_{\pm 0.01}$ | $0.24_{\pm 0.01}$ | $0.12_{\pm 0.00}$ | $0.24_{\pm 0.01}$ | $0.20_{\pm 0.00}$ | $0.19_{\pm nan}$ |
| brier score | $0.46_{\pm 0.01}$ | $0.54_{\pm 0.02}$ | $0.50_{\pm 0.02}$ | $0.54_{\pm 0.03}$ | $0.42_{\pm 0.00}$ | $0.54_{\pm 0.02}$ | $0.47_{\pm 0.01}$ | $0.72_{\pm nan}$ |

Table 19: CIFAR10 – level 4 corruption.

|  | OURS | OURS (RAND) | DROPOUT | DIAG-LAP | ENSEMBLE | MAP | SWAG | VOGN |
|---|---|---|---|---|---|---|---|---|
| LL | $-1.25_{\pm 0.03}$ | $-2.43_{\pm 0.18}$ | $-2.28_{\pm 0.10}$ | $-2.54_{\pm 0.18}$ | $-1.56_{\pm 0.05}$ | $-2.57_{\pm 0.15}$ | $-1.95_{\pm 0.04}$ | $-1.99_{\pm nan}$ |
| error | $0.36_{\pm 0.01}$ | $0.35_{\pm 0.01}$ | $0.35_{\pm 0.01}$ | $0.35_{\pm 0.01}$ | $0.32_{\pm 0.01}$ | $0.35_{\pm 0.01}$ | $0.32_{\pm 0.00}$ | $0.54_{\pm nan}$ |
| ECE | $0.13_{\pm 0.01}$ | $0.27_{\pm 0.01}$ | $0.24_{\pm 0.01}$ | $0.27_{\pm 0.01}$ | $0.14_{\pm 0.01}$ | $0.27_{\pm 0.02}$ | $0.23_{\pm 0.00}$ | $0.22_{\pm nan}$ |
| brier score | $0.51_{\pm 0.02}$ | $0.60_{\pm 0.03}$ | $0.57_{\pm 0.01}$ | $0.61_{\pm 0.02}$ | $0.47_{\pm 0.01}$ | $0.60_{\pm 0.03}$ | $0.53_{\pm 0.00}$ | $0.76_{\pm nan}$ |

Table 20: CIFAR10 – level 5 corruption.

|  | OURS | OURS (RAND) | DROPOUT | DIAG-LAP | ENSEMBLE | MAP | SWAG | VOGN |
|---|---|---|---|---|---|---|---|---|
| LL | $-1.47_{\pm 0.03}$ | $-2.82_{\pm 0.11}$ | $-2.71_{\pm 0.13}$ | $-3.20_{\pm 0.13}$ | $-1.88_{\pm 0.05}$ | $-3.03_{\pm 0.10}$ | $-2.31_{\pm 0.09}$ | $-2.00_{\pm nan}$ |
| error | $0.41_{\pm 0.00}$ | $0.40_{\pm 0.01}$ | $0.40_{\pm 0.01}$ | $0.41_{\pm 0.01}$ | $0.37_{\pm 0.01}$ | $0.40_{\pm 0.00}$ | $0.36_{\pm 0.01}$ | $0.54_{\pm nan}$ |
| ECE | $0.16_{\pm 0.01}$ | $0.31_{\pm 0.01}$ | $0.28_{\pm 0.01}$ | $0.33_{\pm 0.02}$ | $0.17_{\pm 0.01}$ | $0.31_{\pm 0.01}$ | $0.27_{\pm 0.01}$ | $0.19_{\pm nan}$ |
| brier score | $0.58_{\pm 0.00}$ | $0.69_{\pm 0.01}$ | $0.65_{\pm 0.01}$ | $0.72_{\pm 0.03}$ | $0.55_{\pm 0.01}$ | $0.69_{\pm 0.01}$ | $0.61_{\pm 0.01}$ | $0.75_{\pm nan}$ |

## C    EXPERIMENTAL SETUP

### C.1    TOY EXPERIMENTS

We train a single, 2 hidden layer network, with 50 hidden ReLU units per layer using MAP inference until convergence. Specifically, we use SGD with a learning rate of $1 \times 10^{-3}$, momentum of 0.9 and weight decay of $1 \times 10^{-4}$. We use a batch size of 512. The objective we optimise is the Gaussian log-likelihood of our data, where the mean is outputted by the network and the the variance is a hyperparameter learnt jointly with NN parameters by SGD. This variance parameters is shared among all datapoints. Once the network is trained, we perform post-hoc inference on it using different approaches. Since all of these involve the linearized approximation, the mean prediction is the same for all methods. Only their uncertainty estimates vary.

Note that while for this toy example, we could in principle use the full covariance matrix for the purpose of subnetwork selection, we still just use its diagonal (as described in Section 4) for consistency. We use GGN Laplace inference over network weights (not biases) in combination with the linearized predictive distribution in Eq. (8). Thus, all approaches considered share their predictive mean, allowing us to better compare their uncertainty estimates.

All approaches share a single prior precision of $\lambda = 3$. We chose to select the prior precision such that the full covariance approach (optimistic baseline) presents reasonable results. We use the same value for all other methods. We first tried a precision of 1 and found the full covariance approach to produce excessively large errorbars (covering the whole plot). A value of 3 produces more reasonable results.

Final layer inference is performed by computing the full Laplace covariance matrix and discarding all entries except those corresponding to the final layer of the NN. Results for random sub-network selection are obtained with a single sample from a scaled uniform distribution over weight choice.

### C.2    UCI EXPERIMENTS

In this experiment, our fully connected NNs have numbers of hidden layers $h_d = \{1, 2\}$ and hidden layer widths $w_d = \{50, 100\}$. For a dataset with input dimension $i_d$, the number of weights is given by $D = (i_d + 1)w_d + (h_d - 1)w_d^2$. Our 2 hidden layer, 100 hidden unit models have a weight count of the order $10^4$. The non-linearity used is ReLU.

We first obtain a MAP estimate of each model's weights. Specifically, we use SGD with a learning rate of $1 \times 10^{-3}$, momentum of 0.9 and weight decay of $1 \times 10^{-4}$. We use a batch size of 512. The objective we optimise is the Gaussian log-likelihood of our data, where the mean is outputted by the network and the the variance is a hyperparameter learnt jointly with NN parameters by SGD.

For each dataset split, we set aside 15% of the train data as a validation set. We use these for early stopping training. Training runs for a maximum of 2000 epochs but early stops with a patience of 500 if validation performance does not increase. For the larger Protein dataset, these values are 500 and 125. The weight settings which provide best validation performance are kept.

We then perform full network GGN Laplace inference for each model. We also use our proposed Wassertein rule together with the diagonal Hessian assumption to prune every network's weight variances such that the number of variances that remain matches the size of every smaller network under consideration. The prior precision used for these steps is chosen such that the resulting predictor's logliklihood performance on the validation set is maximised. Specifically, we employ a grid search over the values: $\lambda : [0.0001, 0.001, 0.1, 0.5, 1, 2, 5, 10, 100, 1000]$. In all cases, we employ the linearized predictive in Eq. (7). Consequently, networks with the same number of weights make the same mean predictions. Increasing the number of weight variances considered will thus only increase predictive uncertainty.

### C.3    IMAGE EXPERIMENTS

The results shown in Section 5.3 and Appendix B are obtained by training ResNet-18 (and ResNet-50) models using SGD with momentum. For each experiment repetition, we train 7 different models: The first is for: 'MAP', 'Ours', 'Ours (Rand)', 'SWAG', 'Diag-Laplace' and as the first element of

'Ensemble'. We train 4 additional 'Ensemble' elements, 1 network with 'Dropout', and, finally 1 network for 'VOGN'. The methods 'Ours', 'Ours (Rand)', 'SWAG', and 'Diag-Laplace' are applied post training.

For all methods except 'VOGN' we use the following training procedure. The (initial) learning rate, momentum, and weight decay are 0.1, 0.9, and $1 \times 10^{-4}$, respectively. For 'MAP' we use 4 Nvidia P100 GPUs with a total batch size of 2048. For the calculation of the Jacobian in the subnetwork selection phase we use a single P100 GPU with a batch size of 4. For the calculation of the hessian we use a single P100 GPU with a batch size of 2. We train on 1 Nvidia P100 GPU with a batch size of 256 for all other methods. Each dataset is trained for a different number of epochs, shown in Table 21. We decay the learning rate by a factor of 10 at scheduled epochs, also shown in Table 21. Otherwise, all methods and datasets share hyperparameters. These hyperparameter settings are the defaults provided by `PyTorch` for training on ImageNet. We found them to perform well across the board. We report results obtained at the final training epoch. We do not use a separate validation set to determine the best epoch as we found ResNet-18 and ResNet-50 to not overfit with the chosen schedules.

Table 21: Per-dataset training configuration for image experiments.

| DATASET | NO. EPOCHS | LR SCHEDULE |
|---|---|---|
| MNIST | 90 | 40, 70 |
| CIFAR10 | 300 | 150, 225 |

For 'Dropout', we add dropout to the standard ResNet-50 model (He et al., 2016) in between the 2 and 3 convolutions in the bottleneck blocks. This approach follows Zagoruyko & Komodakis (2016) and Ashukha et al. (2020b) who add dropout in-between the two convolutions of a WideResNet-50's basic block. Following Antorán et al. (2020), we choose a dropout probability of 0.1, as they found it to perform better than the value of 0.3 suggested by Ashukha et al. (2020b). We use 16 MC samples for predictions. 'Ensemble' uses 5 elements for prediction. Ensemble elements differ from each other in their initialisation, which is sampled from the He initialisation distribution (He et al., 2015). We do not use adversarial training as, inline with Ashukha et al. (2020b), we do not find it to improve results. For 'VOGN' we use the same procedure and hyper-parameters as used by Osawa et al. (2019) in their CIFAR10 experiments, with the exception that we use a learning rate of $1 \times 10^{-3}$ as we we found a value of $1 \times 10^{-4}$ not to result in convergence. We train on a single Nvidia P100 GPU with a batch size of 256. See the authors' GitHub for more details: `github.com/team-approx-bayes/dl-with-bayes/blob/master/ distributed/classification/configs/cifar10/resnet18_vogn_bs256_ 8gpu.json`.

We modify the standard ResNet-50 and ResNet-18 architectures such that the first $7 \times 7$ convolution is replaced with a $3 \times 3$ convolution. Additionally, we remove the first max-pooling layer. Following Goyal et al. (2017), we zero-initialise the last batch normalisation layer in residual blocks so that they act as identity functions at the start of training.

At test time, we tune the prior precision used for 'Ours', 'Diag-Laplace' and 'SWAG' approximation on a validation set for each approach individually, as in Ritter et al. (2018); Kristiadi et al. (2020). We use a grid search from $1 \times 10^{-4}$ to $1 \times 10^4$ in logarithmic steps, and then a second, finer-grained grid search between the two best performing values (again with logarithmic steps).

## C.4 DATASETS

The 1d toy dataset used in Section 5.1 was taken from Antorán et al. (2020). We obtained it from the authors' github repo: `https://github.com/cambridge-mlg/DUN`. Table 22 summarises the datasets used in Section 5.2. Wine and Protein are available from the UCI dataset repository Dua & Graff (2017). Kin8nm is available from `https://www.openml.org/d/189` Foong et al. (2019b). For the standard splits (Hernández-Lobato & Adams, 2015) 90% of the data is used for training and 10% for validation. For the gap splits (Foong et al., 2019b) a split is obtained per input dimension by ordering points by their values across that dimension and removing the middle 33% of

Table 22: Datasets from tabular regression used in Section 5.2

| Dataset | N Train | N Val (15% train) | N Test | Splits | Output Dim | Output Type | Input Dim | Input Type |
|---------|---------|-------------------|--------|--------|------------|-------------|-----------|------------|
| Wine | 1223 | 216 | 160 | 20 | 1 | Continous | 11 | Continous |
| Wine Gap | 906 | 161 | 532 | 11 | 1 | Continous | 11 | Continous |
| Kin8nm | 6267 | 1106 | 819 | 20 | 1 | Continous | 8 | Continous |
| Kin8nm Gap | 4642 | 820 | 2730 | 8 | 1 | Continous | 8 | Continous |
| Protein | 34983 | 6174 | 4573 | 5 | 1 | Continous | 9 | Continous |
| Protein Gap | 25913 | 4573 | 15244 | 9 | 1 | Continous | 9 | Continous |

the points. These are used for validation. The datasets used for our image experiments are outlined in Table 23.

Table 23: Summary of image datasets. The test and train set sizes are shown in brackets, e.g. (test & train).

| NAME | SIZE | INPUT DIM. | NO. CLASSES | NO. SPLITS |
|------|------|------------|-------------|------------|
| MNIST (LeCun et al., 1998) | 70,000 (60,000 & 10,000) | 784 (28 × 28) | 10 | 2 |
| Fashion-MNIST (Xiao et al., 2017) | 70,000 (60,000 & 10,000) | 784 (28 × 28) | 10 | 2 |
| CIFAR10 (Krizhevsky & Hinton, 2009) | 60,000 (50,000 & 10,000) | 3072 (32 × 32 × 3) | 10 | 2 |
| SVHN (Netzer et al., 2011) | 99,289 (73,257 & 26,032) | 3072 (32 × 32 × 3) | 10 | 2 |

