# OpenReview forum: "Expressive yet Tractable Bayesian Deep Learning via Subnetwork Inference"
_ICLR.cc/2021/Conference — Reject_

### Official Review · AnonReviewer3 · 2020-10-19
**Good PoC but the main experiment raises questions**

**Rating:** 5
**Confidence:** 5

**Review:**

The paper presents a new way for approximating posteriors in Bayesian DNN. The network is split into two subnets. One uses only point estimates while another one uses full (non-diagonal) Gaussian approximation. The structure of that subnet is found by taking largest second derivatives of Hessian of linearized DNN (the authors call it generalized Gauss-Newton (GGN) matrix). The authors show that under very specific conditions such choice correposnds to minimization of Wasserstein-2 distance between their approximation and the true posterior. In the experimental part they provide a set of explorative experiments showing that it may be better to use their approximation for inference in large newtork than both using standard (simple) approximations in large network and full Bayesian inference in small network. This is very nice methodologically and I welcome such demonstration but this can only be considered as a (good) proof of concept. The flagship experiment however looks very inconvincing (see below).

Pros.
1. Interesting idea of finding a better approximation for the posterior on subset of parameters.
2. Methodologically nice PoC.
3. Thorough comparison with alternative similar techniques such as SWAG.

Cons.
1. The authors claim that they theoretically characterize the descrepancy and derive optimal strategy (see contribution 3) but they (0) consider linearized approximation of DNN (they admit this); (1) do this ONLY for regression problem although their flagship experiment is on classification problems; (2) the method they derive is based on un-natural assumption that covariance matrix is diagonal (it the matrix is diagonal there is no way to approximate with a full submatrix anyway). I would not call it an optimal strategy - it rather looks like a reasonalbe heuristic.
2. My major concern is section 5.3. The authors claim that their method estimates uncertainty better than all baselines including deep ensembles (DE). I am afraid in its current form the comparison is not fair. They use only DE of size 5 while their method requires approx. (42K)*(42K) = 1756B of parameters that is enough to keep in memory about 160 initial networks. So it would be fair to compare aganist DE of size 160 since we know that larger DE estimate uncertainties better. It is not that surprizing that the suggested method outperforms other baselines since all of them require much less memory. So I would recomend the authors to compare (1) their current model aganist DE that requires similar amount of memory; (2) their reduced model (that requires approximately the same amount of memory as baselines) aganist other baselines to check wether the proposed algorithm may still estimate uncertainty better given the same memory budget.

---

> ### Author Response · Authors · 2020-11-25
> **Thank you for the feedback!**
>
> We thank the reviewer for their positive comments  and constructive suggestions. We address both individual points below:
>
> ### On the our subnetwork selection strategy
>
> We agree that our phrasing of our subnetwork selection procedure was not the best. **We have revised our claims about subnetwork selection optimality and now simply present our approach as an empirically well-performing subnetwork selection strategy (with theoretical motivation).**
>
> Regarding our approximations:
>
> 1) We employ the Laplace approximation as it is a post-hoc method. It allows us to perform post-hoc subnetwork selection.
>
> 2) The linearized approximation goes hand in hand with the Laplace approximation. The full Hessian is almost always intractable. Instead it is common to resort to the outer product (GGN) approximation. The implied posterior corresponds to that of a linear model and indeed linearisation has been found to perform better for prediction than regular MCMC estimation of the BNN predictive posterior (https://arxiv.org/pdf/1906.11537.pdf).
>
> 3) The diagonal assumption for subnetwork selection is indeed unrealistic. Due to the large size of NN weight spaces, considering the full covariance matrix is always intractable. One of the main findings of this paper is that it is better to perform a cruder approximation when choosing a subnetwork and then perform full covariance inference over that subnetwork than it is to directly use crude factorised approximations over the whole network (see experiments in Sections 5.1 and 5.3).
>
> 4) Concerning your point regarding regression: The GGN approximation is indeed only exact for the linearised model in the regression setting. However it is an accurate approximation for generalised linear models that employ non-linear linking functions, like is the case for classification; see the updated Section 4 (step “1. Point Estimation”) for more justification and details.
>
> **We have adjusted our claims throughout the paper and re-written section 4 in a way that clarifies the points above.**
>
> ### On the computational requirements of subnetwork inference and comparison to baselines
>
> You are correct in that our approach uses more memory than baselines. **We provide additional experiments, as you suggested, showing our method makes an efficient use of parameters:**
>
> Several works have shown that performance of deep ensembles saturates after around a dozen ensemble members, i.e. there are diminishing returns (see e.g. https://arxiv.org/pdf/2007.08483.pdf, or https://arxiv.org/abs/2006.08437 ). We evaluate large deep ensembles (of up to 50+ elements) on our MNIST and CIFAR tasks and replicate this finding (see Figure 5 in Appendix B). We find the performance of subnetwork inference is better than a deep ensemble even for large numbers of ensemble members, in particular for high degrees of dataset shift.
>
> On a separate note, with subnetwork inference, we can choose to use as much compute as we have available. This means that, unlike deep ensembles, our method can be scaled down when on a budget and could potentially improve with more capable hardware. Our experiments are limited by our available hardware: We choose a subnetwork size that allows us to comfortably make predictions with resnet18 on a single p100 GPU (16GB). I.e., while our approach is not explicitly aimed towards memory efficiency, we still do not use excessive amounts of memory. For severely memory-constrained settings (e.g. embedded systems), other approaches might indeed be more viable, as that is not the focus of our work.
>
> We understand that our comparison might be “fairer” if all methods used the exact same memory and compute resources, but we do not think that this is the most useful comparison, as some methods can significantly improve in performance when more memory/compute is available (i.e. ours), while other baselines cannot benefit from having more resources (e.g. deep ensembles, MAP, Dropout, diagonal Laplace), and quickly saturate in their performance.
>
> Finally, this work is aimed to introduce a first viable method to effectively perform inference over subnetworks. We do not claim that our approach solves every issue, but believe that it is an important first step in a promising direction to make Bayesian deep learning more effective. There are many possibilities to make our method more (memory) efficient, which we believe to be beyond the scope of this work, but are certainly keen to explore in future work.

---

### Official Review · AnonReviewer2 · 2020-10-22
**Good idea, solid theoretical foundation, but empirically weak**

**Rating:** 5
**Confidence:** 4

**Review:**

The authors present a new method for Bayesian deep learning motivated by the difficulty of posterior inference in the "overparameterized" regime of deep neural network models. The proposed method provides a principled strategy for selecting a subset of the neural network's parameters (forming a so-called "subnetwork") for which a full-covariance approximate posterior can be computed. The authors use the well-studied Laplace approximation with the generalized Gauss-Newton Hessian approximation for the covariance. An empirical analysis is presented which attempts to assess the efficacy of the proposed method in prediction accuracy and uncertainty quantification.

The presented approach is novel and appears to be a promising contribution to the study of Bayesian neural networks. As a fellow Bayesian, I applaud the authors' efforts. Unfortunately, the paper has a number of significant weaknesses which I detail below. The authors' experimental results appear to me to not sufficiently support some of their claims. There are also a number of formatting issues. As such, I cannot recommend accepting this work in its current state. I would be happy to revisit my rating after revision and discussion.

Pros:
- The basic idea behind the method is very interesting and constructive for the field
- Theoretical justification is solid
- Overall well written with only minor clarity issues

Cons:
- Experimental results lack thoroughness
- Results do not seem to adequately support authors' (somewhat bold) claims
- Lacking detail in certain areas of the method description


I will organize my comments for the authors by section.

#### Section 1
1. "In turn, we can apply more expensive, but more faithful, posterior approximations to just that subnetwork to achieve better uncertainty quantification than if we apply cheaper, but more crude, approximations to the full network."
This sentence is too long and too difficult to read. Please try rephrasing to make it more clear.

#### Section 2

2. It would be nice to see the subnetwork posterior fully defined in terms of the fixed weights, e.g:
$$p(W_s | y,X,W^*) \propto p(y|X,W_s,W^*)p(W_s)p(W^*) = p(W_s|y,X)p(W^*)$$
I don't meant to be pendantic here. It wasn't obvious to me (at first) that the delta functions were, in fact, a degenerate prior over the fixed weights, $W^*$; this would make it more clear.

3. It might be worth discussing here or later what implications the degenerate prior has for the subnetwork posterior. This seems to, at least, preclude any ideas about possibly applying gradient-optimized MAP directly to the subnetwork posterior.
Furthermore, what are the benefits of this approach as opposed to assigning a non-degenerate Gaussian prior with fixed variance to the remaining weights?

#### Section 4

4. "We now analyze the following procedure..."
Are you analyzing the procedure for selecting a subnetwork? Or the entire procedure described previously? This is not made clear.

5. In step 1, you specify the analytical solution for w_MAP, but in section 3 you describe MAP as being performed using stochastic gradient optimization. Which one are you using? Related to previous point, is this the same step as outlined in section 3? Or a separate step?

6. In general, this section needs work. It is not clear to me where you actually detail how the subnetwork is selected. You discuss approximating the optimal subnetwork w.r.t to the Wasserstein distance in equation 11, but this equation requires M_S to be already available. M_S is generated by "a (one-shot) procedure of choice"; I expected this choice to be clearly explained in this section, but this is not the case.

#### Section 5.1

7. It's a bit confusing that you say "50%, 97%, and 99% of model parameters" when really you mean that this percentage of weights were pruned. It would be visually more intuitive as well in Figure 2 if you named these 50%, 3%, 1%, since the posterior size is getting *smaller* with each one.

8. Just to clarify, homeoscedastic here is w.r.t to the sequence of data points? i.e. you have one variance rather than one per data point?

9. Please provide some explanation for the prior precision being set to 3.

10. Bolding most of the last sentence in section 5.1 is unnecessary and looks weird. I would suggest bolding individual words or phrases in the sentence, or just not bolding at all.

#### Section 5.2

11. 1e4 ->$10^4$

12. It would be helpful to provide a very brief summary of the three datasets as well as what the "gap variants" are (it's fairly straightforward, just creating "gaps" in the training data). I cannot find the "kin8nm" dataset on UCI, so at bare minimum this needs to be clarified.

13. Figure 3 is quite difficult to interpret at first glance. You should at least use a discretized perceptually uniform colormap to make more visually apparent the pattern in increasing network size. I would also consider using a line plot rather than scatter plot here, but this is a matter of taste.

14. Why are you using the log likelihood as opposed to the full posterior probability? Since we're in a Bayesian setting here, it seems worth considering the priors.

15. While log likelihood can, in principle, serve as an indirect proxy for uncertainty calibration, there are important caveats to consider which make it unconvincing as a standalone measure (see "Pitfalls of In-Domain Uncertainty Estimation...", Ashukha et al. 2020). Furthermore, a quick scan of recent literature (cited in this work) confirms that most authors tend to use multiple methods of assessment (e.g. Brier score, accuracy calibration curves, etc), not just log likelihood. The raw likelihood scores also don't provide any interpretability in *how much better* one model is than another, i.e. what concrete impact does the difference have on uncertainty quantification? Thus, **it's unconvincing that here (and elsewhere) you rely entirely on log ilkelihood** to demonstrate the alleged superiority of your method.

16. It's noteworthy that 2/3 of the "gap" datasets produce inconsistent results, particularly considering that they are designed specifically to test out-of-sample uncertainty. This needs to be discussed.

17. typo: modelling -> modeling

18. In light of points 15 and 16, I do not think you have sufficient empirical basis to make the claim that "...it is better to perform subnetwork inference in a large model than full network inference in a small one". The results are simply too weak to support this. I suggest either running a more comprehensive experiment or dialing back the confidence of this claim.

#### Section 5.3

19. You mention "grid search" multiple times but do not provide any indication of the grid you searched over.

20. typo: sitribution -> distribution

21. Please specify the error metrics being used in figure 4 (and discussed in the text).

22. Once again, marginally higher likelihood scores are not, in my view, sufficient basis to claim that your method assigns "high uncertainty to out of distribution points" or that it is better calibrated. This is especially the case considering that there seems to be little to no improvement in the error. The case for improved "robustness" also appears to be equally unfounded on these grounds.

---

> ### Author Response · Authors · 2020-11-25
> **Thank you for the feedback! (1/2)**
>
> We thank the reviewer for their thorough feedback and constructive suggestions. We address your individual points below:
>
> **1-2**
> **Agreed; we rephrased and clarified in the updated text.**
>
> **3**
>
> That is a good question: we could indeed take the square of each weight’s gradients to fill the diagonal of our covariance matrix.
> We did not do this due to additional implementation complications: having larger matrices filled with zeros would force us to use sparse matrix optimised operations to retain computational tractability. We do not see this additional complication as necessary considering the poor performance of factorised posterior approximations in neural networks (https://proceedings.neurips.cc/paper/2020/hash/b6dfd41875bc090bd31d0b1740eb5b1b-Abstract.html).  In practise, our main focus is on capturing correlation among weights. Indeed, in our experiments we consistently find full covariance subnetwork inference to handily outperform diagonal covariance approaches.
>
> **4-6**
>
> **We have significantly revised Section 4 for clarity.** In the updated version we focus on a generalised linear model where the MAP setting is found via gradient descent. This matches the setting used in all our experiments. Note that step 2 merely defines a general mask that can be produced by any pruning method; we then later describe how to choose that mask.
>
> **7**
>
> **We agree. The change has been implemented.**
>
> **8**
>
> Yes, that is correct. We believe this is decently standard nomenclature for predictive models.
>
>
> **9**
>
> Results for toy 1d regression are mostly qualitative. As such, it is hard to come up with a rigorous way of selecting hyperparameters. We choose to select the prior precision such that the full covariance approach (optimistic baseline) presented reasonable results and used the same value for all other methods. We first tried a precision of 1 and found the full covariance approach to produce excessively large errorbars (covering the whole plot). We then tried a value of 3 and found the result to look reasonable. **We describe this in our new implementation detail appendix.**
>
> **11**
>
> **Thanks, Done!**
>
> **12**
>
> Indeed Kin8nm is not a UCI dataset. Good catch!. **We added dataset descriptions and locations in the implementation detail appendix.**
>
> **14**
>
> We are a bit confused by your suggestion. We present test data log-likelihoods to estimate the predictive performance of different methods. It is unclear to us how a prior would come into play here.
>
> **15**
>
> We would like to clarify that the experiments in section 5.2 focus on regression tasks. We are not aware of any widespread (within the ML community) uncertainty calibration metrics for regression apart from LL. Indeed, most other paper that use the datasets we consider provide results in terms of LL (https://arxiv.org/abs/1502.05336, https://papers.nips.cc/paper/2017/file/9ef2ed4b7fd2c810847ffa5fa85bce38-Paper.pdf https://arxiv.org/abs/1506.02142 https://arxiv.org/abs/1811.09385 ). The papers you reference only address classification settings. **For our image classification experiments (Sec 5.3), we do provide ECE and Brier score plots in the appendix (see Figure 7 in Appendix B).**
>
> **16-18**
>
> In two gap datasets you mention, the predictions from the MAP estimated single hidden layer networks are superior to those from the larger models. In linearized NNs, bayesian inference does not alter the predicted mean. This means that, on the aforementioned datasets, small networks will have more accurate mean predictions when performing subnetwork inference. Gap datasets are especially designed to be prone to models overfitting on them, to which we attribute smaller networks being more accurate. Having said this, even on these datasets, larger models see a larger increase in LL than smaller ones when performing inference over an equal number of weights. This is consistent with the results on our other datasets. Given the popularisation of large NNs in the past few years, and their strong empirical performance on real world data, we consider that it is indeed generally better to use a large model and perform subnetwork inference than performing full network inference on a small NN. **After reading your comments we acknowledge that this claim may be too aggressive and have relaxed it to: “Given the same amount of compute, larger models benefit more from subnetwork inference than small ones.”**
>
> **19**
>
> Agreed! We first performed a grid search from 1e-4 to 1e4 in logarithmic steps, and then a second, finer-grained grid search between the two best performing values. Following Ritter et al. 2018, Kristiadi et al. 2020,  we perform this search after training, using a validation set. Similarly to those previous works, we found tuning the prior precision to empirically improve results. **Full descriptions are in the new implementation detail appendix.**

---

> > ### Author Response · Authors · 2020-11-25
> > **Thank you for the feedback! (2/2)**
> >
> > **20**
> >
> > Thanks, fixed that!
> >
> > **21**
> >
> > It is simply classification error: $\frac{\text{N incorrect}}{Total}$. **We clarify this in the updated manuscript.**
> >
> > **22**
> >
> > improved "robustness" also appears to be equally unfounded on these grounds.
> > We believe there may have been some miscommunication with regards to the term “calibration”. We take calibration to refer to a model placing more uncertainty on points on which it is more likely to make wrong predictions (http://www.gatsby.ucl.ac.uk/~balaji/ensembles_nipsbdl16.pdf). We don’t think that the likelihood obtained by subnetwork inference are just “marginally higher”, but significantly higher. The predictive error is similar, so any improvement in log-likelihood must be due to better uncertainty calibration. **We furthermore provide Brier score and ECE results in the appendix, which are widely used measures for calibration; these also support our claims that our method is well-calibrated, especially for increasing amounts of dataset shift (see Figure 7 in Appendix B).**

---

### Official Review · AnonReviewer1 · 2020-10-28
**Interesting ideas but various clarifications are needed**

**Rating:** 6
**Confidence:** 4

**Review:**

The paper proposes to approximate the posterior distribution of a Bayesian neural network by an approximation that consists of a deterministic component. The authors select a sub network and infer approximate posterior distributions over the weights in the sub network. All other weights are estimated via MAP point estimation.  A sufficiently small sub-network allows high fidelity posterior approximations that do not make restrictive mean field assumptions to be tractable.

The paper is generally well written and easy to follow. The idea that BNNs have too many parameters for reliably inferring posterior distributions over all of them is a reasonable one. Splitting the posterior approximation into a deterministic and stochastic component to deal with this issue is interesting.  The experiments do indicate improvements over fully factorized posterior approximations.

Concerns:
* The process of selecting the sub-network over which the full posterior distribution is inferred is crucial and herein lies my main concern with the approach presented in the paper. Minimizing the Wasserstein distance between the subnetwork posterior p(W_s | data) and the true posterior over all weights p(W | data) is sensible. However, since the true posterior is intractable, the authors instead appear to minimize the distance to an Laplace approximated posterior q(W | data). While this is fine for smaller models when the approximation can use a full covariance,  why does it make sense for larger models where the authors use diagonal approximations to the generalized Gauss Newton matrix (Section 5.3)? If the goal is to do better than such diagonal approximations then why treat the subnetwork with the lowest Wasserstein distance to such crude approximations as optimal? How much worse is the  random selection strategy on the tasks listed in section 5.3?
* In the toy experiments, when the full covariance posteriors approximations  are available, using the deterministic + stochastic sub-network approximation does a little bit worse than the full covariance approximation but better than the diagonal approximation. If instead the diagonal posterior approximation is used to select the subnetwork in these models, does the resulting approximation still improve upon the diagonal approximation (as seems to be happening in the experiments in 5.3)?
* The experiment in section 5.3 has other curious issues. Different methods appear to be using different priors (Gaussians with different precisions). How do we then know that the benefits reported stem from the proposed approximation rather than the differences in the model, especially since diagonal Laplace is using a Gaussian prior with a precision that is 80 times smaller?  What priors were used for deep ensembles and SWAG?
* The grid search by which the prior precisions were selected needs more details. The presented numbers are surprisingly small suggesting that a-priori  with high probability we expect all weights to be zero, and hence the prior predictive functions to be all zero as well. This does not seem like a sensible prior.
* At the very least there needs to be a discussion about sparse Bayesian deep learning techniques (and preferably an empirical comparison)[1, 2, 3], that use sparsity inducing priors to prune away weights and nodes from a larger network instead of the approach presented here.
* (Minor) The conclusion of 5.3 that subnetwork posteriors are better calibrated are not supported by Figure 4. I would suggest moving the ECE and Brier score plots from the appendix to Figure 4.

Overall, although I have several concerns they primarily stem from experimental issues in 5.3. Assuming that the authors are able to sufficiently address these in the rebuttal, I am leaning towards a borderline accept.

[1] https://www.jmlr.org/papers/v20/19-236.html

[2] https://papers.nips.cc/paper/7372-posterior-concentration-for-sparse-deep-learning.pdf

[3] https://arxiv.org/pdf/2002.10243.pdf

---

> ### Author Response · Authors · 2020-11-25
> **Thank you for the feedback! (1/2)**
>
> Dear AnonReviewer1,
>
> We thank you for your helpful feedback! It has helped us make our paper stronger and more clear. We hope that our updated draft alleviates your concerns with our work. We address individual points below:
>
> ### On the diagonal approximation for subnetwork selection
>
> You bring up a good point: we estimate a full covariance posterior over a subset of weights but the subset is chosen by making a diagonal approximation. Due to the large parameter space of BNNs, it is necessary to use a crude approximation somewhere. Capturing all weight correlations is simply intractable. Our paper proposes to not use a crude approximation for inference, but instead use a crude approximation for subnetwork choice. We  then do expressive inference over that subnetwork. In practice we find this to significantly outperform a direct diagonal posterior approximation (See Sections 5.1, 5.3).
> **We have revised our claims about subnetwork selection optimality and now simply present our approach as an empirically well-performing subnetwork selection strategy (with theoretical motivation).**
>
> We appreciate your concerns regarding the faithfulness of diagonal subnetwork selection. We conduct the experiment you suggest, comparing our approach to random selection on MNIST rotations and CIFAR corruptions. Empirically, we find random subnetwork selection to perform poorly (similar to MAP) on ResNet18. It performs much worse than on the toy data in Section 5.1. We hypothesise this is because it is substantially more difficult to randomly find a good subnetwork in a large model such as a ResNet18, than it is in a small fully-connected network. **We have added the random baseline to the updated manuscript (see “Ours (Rand)” in Figure 4).**
>
> ### Diagonal or Full Approach to subnetwork selection in toy experiments
>
> We in fact always consider a diagonal approximation when selecting subnetworks, even in the 1D toy regression example where the full GGN is tractable. Thus all sections (including the toy experiments) use diagonal subnetwork selection. **We have updated our prose to clarify this.**
>
> Following from our previous answer: our toy experiments aim to show that making a diagonal approximation for subnetwork selection, but then using a full-covariance approach for inference (“Wass” in 5.1) is much better than using a diagonal approximation directly for inference (“Diag” in 5.1). An important conclusion from our work is that using crude approximations for subnetwork selection in combination with expressive approximations for inference is significantly better than using crude approximations for inference directly (Sections 5.1, 5.3). **We have more strongly emphasised this point.**
>
> ### On our choice of priors for the Image experiments (Sec 5.3)
>
> We would like to clarify that all models (including deep ensembles and SWAG) use the same prior precision during training (i.e. the standard weight decay of 1e-4 for ResNet18) for comparability, e.g. all trained models are identical. In fact, for each experiment repetition, we only trained 6 different models: The first is used for the results in: “MAP”, “Ours”, “SWAG”, “Diag-Laplace” and as the first element of “Ensemble.” We trained additional “Ensemble” elements and, finally, 1 network with “Dropout”.
>
> At test time, we tune the prior precision used for the Laplace approximation on a validation set for each approach individually, as done in (Ritter et al. 2018, Kristiadi et al. 2020). We use a grid search approach. Although this results in “Ours”, “Diag-Laplace” and “SWAG” using different priors, we believe that is the best setting for comparison as we present each method in its strongest configuration.
> **Our updated text includes an additional appendix on our experimental setup to clarify points like these and make our results easier to reproduce.**

---

> > ### Author Response · Authors · 2020-11-25
> > **Thank you for the feedback! (2/2)**
> >
> > ### Details on our prior precision grid searches
> >
> > We first performed a grid search from 1e-4 to 1e4 in logarithmic steps, and then a second, finer-grained grid search between the two best performing values. Following Ritter et al. 2018, Kristiadi et al. 2020,  we perform this search after training, using a validation set. Similarly to those previous works, we found tuning the prior precision to empirically improve results.
> >
> > With regards to the values found. We acknowledge that they might seem large. However it is difficult to gain intuition about the implications of prior variance specification in the high-dimensional weight space of a NN. Some possible explanations 1) In very large covariance matrices large jitter values can be needed to ensure positive semidefiniteness. 2) Large overparameterized NNs are underspecified by the data and require strong prior constraints to produce reasonably bounded uncertainty estimates.
> >
> > We also note that, unlike Ritter et al. 2018, who scale their Hessian matrix to obtain reasonable results, we do not need to do this. We attribute this to our use of the linearised predictive function, where there is no mismatch between GGN the model and posterior.
> >
> > ### Relation to sparse Bayesian deep learning
> >
> > Indeed subnetwork inference is related to more traditional sparse Bayesian methods in deep learning.  **As such, we added a discussion of such approaches in the related work section.**
> >
> > We did not include them initially, since the end goal is different. Generally speaking, the papers you cited use Bayesian reasoning to perform model selection. On the other hand, we aim to make inference tractable in a fixed model.
> > It is unclear to us how we would perform a fair comparison with the methods you mention. We do not actually compress or prune weights. However, we estimate (co-)variances over only a subset of the weights. Importantly, this means that our approach retains the full predictive power of the full network to retain high predictive accuracy.
> >
> > ### Moving ECE and Brier to the main text
> >
> > Thanks for the suggestion. Indeed, subnetwork inference delivers strong calibration performance. Unfortunately we could not fit the results in the main text. They are in the appendix for now (see Figure 7 in Appendix B) but we will try to move them for the camera ready version.

---

### Official Review · AnonReviewer4 · 2020-10-28

**Rating:** 6
**Confidence:** 4

**Review:**

#### Summary

The authors focus on the important problem of scalable approximate inference in Bayesian NNs. More specifically, they propose a method for scalable BNNs via a (full-covariance Gaussian) Laplace approximation on a (Wasserstein-based) pruned subnetwork within a deterministically-trained model. They include a theoretical analysis for a simple generalized linear model, and experiments on 1D regression, tabular regression, and larger-scale image classification with CIFAR-10 (using the dataset shift setup from Ovadia et al., (2019)). From the experiments, they show that their method generally outperforms comparable methods (including deep ensembles) on metric performance and on the ability to capture in-between uncertainty.

#### Strengths

- Scalable approximate inference for Bayesian models is an important research area.
- Expressive, diverse approximate posteriors is also an important area of research, especially given the limitations of mean-field VI and recent literature.
- The proposed method demonstrates better results for robustness to dataset shifts, as well as in-between uncertainty that mean-field VI misses.
- In general, the paper is well-written, clearly-motivated, includes both theoretical and empirical results, and adequately compares to, or discusses, relevant literature in the space.

#### Weaknesses

- The authors push on the idea of *scalable* approximate inference, yet the largest experiment shown is on CIFAR-10. Given this focus on scalability, and the experiments in recent literature in this space, I think experiments on ImageNet would greatly strengthen the paper (though I sympathize with the idea that this can a high bar from a resources standpoint).
- As I noted down below, the experiments currently lack results for the standard variational BNN with mean-field Gaussians. More generally, I think it would be great to include the remaining models from Ovadia et al. (2019). More recent results from ICML could also useful to include (as referenced in the related works sections).

#### Recommendation

Overall, I believe this is a good paper, but the current lack of experiments on a dataset larger than CIFAR-10, while also focusing on scalability, make it somewhat difficult to fully recommend acceptance. Therefore, I am currently recommending marginal acceptance for this paper.

#### Additional comments

- p. 5-7: Including tables of results for each experiment (containing NLL, ECE, accuracy, etc.) in the main text would be helpful to more easily assess
- p. 7: For the MNIST experiments, in Ovadia et al. (2019) they found that variational BNNs (SVI) outperformed all other methods (including deep ensembles) on all shifted and OOD experiments. How does your proposed method compare? I think this would be an interesting experiment to include, especially since the consensus in Ovadia et al. (2019) (and other related literature) is that full variational BNNs are quite promising but generally methodologically difficult to scale to large problems, with relative performance degrading even on CIFAR-10.

##### Minor
- p. 6: In the phrase "for 'in-between' uncertainty", the first quotation mark on 'in-between' needs to be the forward mark rather than the backward mark (i.e., $`in-between'$).
- p. 7: s/out of sitribution/out of distribution/
- p. 8: s/expensive approaches 2) allows/expensive approaches, 2) allows/
- p. 8: s/estimates 3) is/estimates, and 3) is/
- In the references:
  - Various words in many of the references need capitalization, such as "ai" in Amodei et al. (2016), "bayesian" in many of the papers, and "Advances in neural information processing systems" in several of the papers.
  - Dusenberry et al. (2020) was published in ICML 2020
  - Osawa et al. (2019) was published in NeurIPS 2019
  - Swiatkowski et al. (2020) was published in ICML 2020
- p. 13, supplement, Fig. 5: error bar regions should be upper and lowered bounded by [0, 1] for accuracy.
- p. 13, Table 2: Splitting this into two tables, one for MNIST and one for CIFAR-10, would be easier to read.

---

> ### Author Response · Authors · 2020-11-25
> **Thank you for the feedback!**
>
> Thank you very much for your kind words and helpful feedback! We address individual points below:
>
>
> ### Scaling to larger datasets
>
> You bring up a good point. Our approach is scalable in the size of the weights as it decouples network size from the dimensionality of the space over which inference is performed. Therefore, we would argue that our validation in terms of scalability is provided by the size of the network we use. **In the appendix (see Figure 8 in Appendix B), we provide additional results on MNIST image classification with ResNet50 (as opposed to ResNet18 in the original version of the paper) to strengthen our claim.** In principle, the cost of subnetwork inference should scale linearly with the number of training samples, as it just requires summing more Jacobian outer products. The memory consumption is constant in the dataset size.  **We have adjusted the claims in our paper to prevent confusion regarding this.**
>
> ### MFVI and other baselines
>
> **We added mean-field VI BNNs to our image classification experiments in the revised manuscript.** In particular, we implemented VOGN, a natural gradient version of MFVI that scales well to large networks (https://arxiv.org/abs/1906.02506). We use the hyperparameter settings provided by the authors. We find that VOGN tends to underfit both datasets, although the issue is larger for CIFAR10. Despite this, similarly to Ovadia et. al. (2019), we find MFVI to provide large uncertainties for rotated MNIST digits. Here it performs similarly to subnetwork inference. However, this is not the case for CIFAR10 corruptions, where MFVI fares poorly both in terms of accuracy and uncertainty estimation.
>
> Note that last-layer approaches performed poorly in Ovadia et al. (2019), which is why we considered these to be less interesting/important to include. By “more recent results from ICML [...] (as referenced in the related work section)”, we believe you are referring to Dusenberry et al. (2020) and Swiatkowski et al. (2020)? We did not compare to Swiatkowski et al. (2020), as they propose a more efficient parameterization of mean-field Gaussian VI (i.e. an approximation that is less expressive than mean-field), which is not expected to perform better than the mean-field Gaussian methods we assessed (i.e. diagonal Laplace and VOGN). Similarly, Dusenberry et al. (2020) also focus on efficiency, and as a result, their experimental results show that their approach performs consistently worse than deep ensembles. We focus on increased expressivity as opposed to increased efficiency.
>
> ### NLL, ECE, accuracy tables
>
> We agree that this would be useful, but unfortunately the space constraints prevent us from moving the tables to the main text. They can all be found in Appendix B.
>
> ### Minor suggestions / typos
>
> **Thanks a lot for all these suggestions, which we incorporated into the revised paper!**

---

### Author Response · Authors · 2020-11-25
**Paper update overview**

We thank the reviewers for their time in reading our paper, insightful comments and helpful suggestions. We apologise for the late response. We have taken this time to implement your suggestions into our manuscript, resulting in the following **significant changes:**

* We have added an additional experiment comparing our method to deep ensembles with a much larger number of ensemble members as previously (i.e. 50+ instead of 5), and studying how the number of ensemble members affects performance (see Figure 5 in Appendix B).
* We have added image classification results for VOGN, a scalable, state-of-the-art approach for mean-field variational inference in DNNs (Osawa et al., 2019). See Figure 4.
* We have added image classification results with ResNet50 (as opposed to ResNet18) to strengthen our claim about scalability in the number of model parameters (see Figure 8 in Appendix B).
* We have added subnetwork inference with random subnetwork selection as a baseline to our image classification experiments in Section 5.3; see baseline “Ours (Rand)” in Fig. 4.
* We have significantly revised Section 4. We have generalized our theory to generalized linear models (i.e. including both regression and classification) learnt by gradient descent (as used in our experiments), and better justified why the Laplace approximation is a faithful approximation to the true posterior for classification. We have also clarified the subnetwork selection procedure.
* We have revised our claims about subnetwork selection optimality and now instead present our approach as an empirically well-performing subnetwork selection strategy (with theoretically grounded motivation).
* We rephrased any references to our approach as a “pruning” method to instead a “subnetwork selection” method for clarity (as we do not do weight pruning in the classical sense). This involved updating Fig. 2 to state the percentage of weights retained instead of those “pruned”.
* We have added a section describing our experimental setup and the datasets we use in detail; see Appendix C.
* We have included a discussion on sparse Bayesian Deep Learning methods in the related work (see end of “Inference over Subspaces” paragraph in Section 6).
* We have relaxed the claim from experimental section 5.2. It now reads: “Given the same amount of compute, larger models benefit more from subnetwork inference than small ones.”
* We re-made Figure 1 to be more legible and visually appealing.
* We have added significantly more detail to the explanation of the linearized Laplace approximation in Section 3, Step #3, in order to improve clarity and intuition.
* We have done a number of updates to phrasing to enhance clarity and language error corrections throughout the paper as described in individual reviewer responses below.

Please refer to our individual responses to all reviewers for further clarifications.

---

### Decision · Program_Chairs · 2021-01-07
**Final Decision**

**Decision:**

Reject

**Comment:**

This paper propose an approach to efficient Bayesian deep learning by applying Laplace approximations to sub-structures within a larger network architecture. In terms of strengths, scalable approximate Bayesian inference methods for deep learning models are an important and timely topic. The paper includes an extensive set of experiments with promising results.

In terms of issues, the reviewers originally raised many concerns and the authors provided a large update to the paper. However, following that update and the discussion, several concerns remain. First, the reviewers noted that the originally submitted draft made claims about the optimality of the sub-network selection procedure that were incorrect due to the use of a diagonal approximation. The authors subsequently retracted these claims and re-focused on the idea that the subset selection approach is theoretically well-motivated heuristic that performs well empirically. Following the discussion, the reviewers continued to express concerns about the heuristic nature of this procedure.

A second point has to do with scalability. The reviewers noted that the authors had only evaluated their approach on small data sets, leaving open the question of how scalable the method is. The authors responded by adding experiments on the same data sets using larger models, which does not squarely address the issue raised. Third, an additional point was raised regarding the lack of control of resource use in the experiments. The authors note that their approach can use more resources when available while many other methods can not. However, some methods including deep ensembles can also expand to use more resources, as can posterior ensembles produced using MCMC methods like SGLD and SGHMC. The authors need to consider quantifying space-performance and time-performance trade-offs in the same units for different approaches to satisfactorily address this issue. While the authors added one set of experiments looking at deep ensembles in isolation, their conclusions that performance saturates for these models at low ensembles sizes seems to be hasty in some cases (e.g., deep ensembles show continued improvement for large corruptions in Figure 5(right) despite the claim by the authors that the models saturate after 15 epochs).

In summary, this appears to be a promising approach. While the authors made significant efforts to correct issues and address questions with the original draft, the majority view of the reviewers following discussion is that this paper requires additional work to more carefully expand on the revised results and to address the heuristic status of the sub-network selection approach.